# PHASE TRANSITIONS FOR THE INFORMATION BOTTLE-NECK IN REPRESENTATION LEARNING

**Tailin Wu**
Stanford
tailin@cs.stanford.edu

**Ian Fischer**
Google Research
iansf@google.com

## ABSTRACT

In the Information Bottleneck (IB), when tuning the relative strength between compression and prediction terms, how do the two terms behave, and what's their relationship with the dataset and the learned representation? In this paper, we set out to answer these questions by studying multiple phase transitions in the IB objective: $\text{IB}_\beta[p(z|x)] = I(X;Z) - \beta I(Y;Z)$ defined on the encoding distribution $p(z|x)$ for input $X$, target $Y$ and representation $Z$, where sudden jumps of $\frac{dI(Y;Z)}{d\beta}$ and prediction accuracy are observed with increasing $\beta$. We introduce a definition for IB phase transitions as a qualitative change of the IB loss landscape, and show that the transitions correspond to the onset of learning new classes. Using second-order calculus of variations, we derive a formula that provides a practical condition for IB phase transitions, and draw its connection with the Fisher information matrix for parameterized models. We provide two perspectives to understand the formula, revealing that each IB phase transition is finding a component of maximum (nonlinear) correlation between $X$ and $Y$ orthogonal to the learned representation, in close analogy with canonical-correlation analysis (CCA) in linear settings. Based on the theory, we present an algorithm for discovering phase transition points. Finally, we verify that our theory and algorithm accurately predict phase transitions in categorical datasets, predict the onset of learning new classes and class difficulty in MNIST, and predict prominent phase transitions in CIFAR10.

## 1 INTRODUCTION

The Information Bottleneck (IB) objective (Tishby et al., 2000):

$$\text{IB}_\beta[p(z|x)] := I(X;Z) - \beta I(Y;Z) \tag{1}$$

explicitly trades off model compression ($I(X;Z)$, $I(\cdot;\cdot)$ denoting mutual information) with predictive performance ($I(Y;Z)$) using the Lagrange multiplier $\beta$, where $X, Y$ are observed random variables, and $Z$ is a learned representation of $X$. The IB method has proved effective in a variety of scenarios, including improving the robustness against adversarial attacks (Alemi et al., 2016; Fischer, 2018), learning invariant and disentangled representations (Achille & Soatto, 2018a;b), underlying information-based geometric clustering (Strouse & Schwab, 2017b), improving the training and performance in adversarial learning (Peng et al., 2018), and facilitating skill discovery (Sharma et al., 2019) and learning goal-conditioned policy (Goyal et al., 2019) in reinforcement learning.

From Eq. (1) we see that when $\beta \to 0$ it will encourage $I(X;Z) = 0$ which leads to a trivial representation $Z$ that is independent of $X$, while when $\beta \to +\infty$, it reduces to a maximum likelihood objective[1] that does not constrain the information flow. Between these two extremes, how will the IB objective behave? Will prediction and compression performance change smoothly, or do there exist interesting transitions in between? In Wu et al. (2019), the authors observe and study the learnability transition, i.e. the $\beta$ value such that the IB objective transitions from a trivial global minimum to learning a nontrivial representation. They also show how this first phase transition relates to the structure of the dataset. However, to answer the full question, we need to consider the full range of $\beta$.

---

[1]For example, in classification, it reduces to cross-entropy loss.

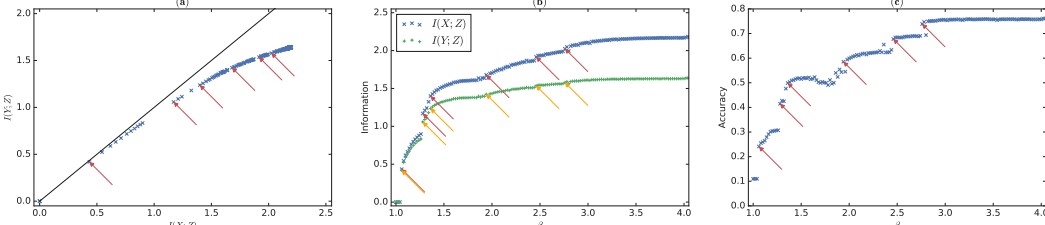

Figure 1: CIFAR10 plots **(a)** showing the information plane, as well as $\beta$ vs **(b)** $I(X;Z)$ and $I(Y;Z)$, and **(c)** accuracy, all on the training set with 20% label noise. The arrows point to empirically-observed phase transitions. The vertical lines correspond to phase transitions found with Alg. 1.

**Motivation.**    To get a sense of how $I(Y;Z)$ and $I(X;Z)$ vary with $\beta$, we train Variational Information Bottleneck (VIB) models (Alemi et al., 2016) on the CIFAR10 dataset (Krizhevsky & Hinton, 2009), where each experiment is at a different $\beta$ and random initialization of the model. Fig. 1 shows the $I(X;Z)$, $I(Y;Z)$ and accuracy vs. $\beta$, as well as $I(Y;Z)$ vs. $I(X;Z)$ for CIFAR10 with 20% label noise (see Appendix I for details).

From Fig. 1**(b)(c)**, we see that as we increase $\beta$, instead of going up smoothly, both $I(X;Z)$ and $I(Y;Z)$ show multiple phase transitions, where the slopes $\frac{dI(X;Z)}{d\beta}$ and $\frac{dI(Y;Z)}{d\beta}$ are discontinuous and the accuracy has discrete jumps. The observation lets us refine our question: When do the phase transitions occur, and how do they depend on the structure of the dataset? These questions are important, since answering them will help us gain a better understanding of the IB objective and its close interplay with the dataset and the learned representation.

Moreover, the IB objective belongs to a general form of two-term trade-offs in many machine learning objectives: $L = \text{Prediction-loss} + \beta \cdot \text{Complexity}$, where the complexity term generally takes the form of regularization. Usually, learning is set at a specific $\beta$. Many more insights can be gained if we understand the behavior of the prediction loss and model complexity with varying $\beta$, and how they depend on the dataset. The techniques developed to address the question in the IB setting may also help us understand the two-term tradeoff in other learning objectives.

**Contributions.**    In this work, we begin to address the above question in IB settings. Specifically:

- We identify a *qualitative* change of the IB loss landscape w.r.t. $p(z|x)$ for varying $\beta$ as IB phase transitions (Section 3).
- Based on the definition, we introduce a quantity $G[p(z|x)]$ and use it to prove a theorem giving a practical condition for IB phase transitions. We further reveal the connection between $G[p(z|x)]$ and the Fisher information matrix when $p(z|x)$ is parameterized by $\boldsymbol{\theta}$ (Section 3).
- We reveal the close interplay between the IB objective, the dataset and the learned representation, by showing that in IB, each phase transition corresponds to learning a new nonlinear component of maximum correlation between $X$ and $Y$, orthogonal to the previously-learned $Z$, and each with decreasing strength (Section 4).

To the best of our knowledge, our work provides the first theoretical formula to address IB phase transitions in the most general setting. In addition, we present an algorithm for iteratively finding the IB phase transition points (Section 5). We show that our theory and algorithm give tight matches with the observed phase transitions in categorical datasets, predict the onset of learning new classes and class difficulty in MNIST, and predict prominent transitions in CIFAR10 experiments (Section 6).

## 2    RELATED WORK

The Information Bottleneck Method (Tishby et al., 2000) provides a tabular method based on the Blahut-Arimoto (BA) Algorithm (Blahut, 1972) to numerically solve the IB functional for the optimal encoder distribution $P(Z|X)$, given the trade-off parameter $\beta$ and the cardinality of the representation

variable $Z$. This work has been extended in a variety of directions, including to the case where all three variables $X, Y, Z$ are multivariate Gaussians (Chechik et al., 2005), cases of variational bounds on the IB and related functionals for amortized learning (Alemi et al., 2016; Achille & Soatto, 2018a; Fischer, 2018), and a more generalized interpretation of the constraint on model complexity as a Kolmogorov Structure Function (Achille et al., 2018). Previous theoretical analyses of IB include Rey & Roth (2012), which looks at IB through the lens of copula functions, and Shamir et al. (2010), which starts to tackle the question of how to bound generalization with IB. We will make practical use of the original IB algorithm, as well as the amortized bounds of the Variational Informormation Bottleneck (Alemi et al., 2016) and the Conditional Entropy Bottleneck (Fischer, 2018).

Phase transitions, where key quantities change discontinuously with varying relative strength in the two-term trade-off, have been observed in many different learning domains, for multiple learning objectives. In Rezende & Viola (2018), the authors observe phase transitions in the latent representation of $\beta$-VAE for varying $\beta$. Strouse & Schwab (2017b) utilize the kink angle of the phase transitions in the Deterministic Information Bottleneck (DIB) (Strouse & Schwab, 2017a) to determine the optimal number of clusters for geometric clustering. Tegmark & Wu (2019) explicitly considers critical points in binary classification tasks using a discrete information bottleneck with a non-convex Pareto-optimal frontier. In Achille & Soatto (2018a), the authors observe a transition on the tradeoff of $I(\theta; X, Y)$ vs. $H(Y|X, \theta)$ in InfoDropout. Under IB settings, Chechik et al. (2005) study the Gaussian Information Bottleneck, and analytically solve the critical values $\beta_i^c = \frac{1}{1-\lambda_i}$, where $\lambda_i$ are eigenvalues of the matrix $\Sigma_{x|y}\Sigma_x^{-1}$, and $\Sigma_x$ is the covariance matrix. This work provides valuable insights for IB, but is limited to the special case that $X$, $Y$ and $Z$ are jointly Gaussian. Phase transitions in the general IB setting have also been observed, which Tishby (2018) describes as "information bifurcation". In Wu et al. (2019), the authors study the first phase transition, i.e. the learnability phase transition, and provide insights on how the learnability depends on the dataset. Our work is the first work that addresses all the IB phase transitions in the most general setting, and provides theoretical insights on the interplay between the IB objective, its phase transitions, the dataset, and the learned representation.

## 3 FORMULA FOR IB PHASE TRANSITIONS

### 3.1 DEFINITIONS

Let $X \in \mathcal{X}, Y \in \mathcal{Y}, Z \in \mathcal{Z}$ be random variables denoting the input, target and representation, respectively, having a joint probability distribution $p(X, Y, Z)$, with $\mathcal{X} \times \mathcal{Y} \times \mathcal{Z}$ its support. $X, Y$ and $Z$ satisfy the Markov chain $Z - X - Y$, i.e. $Y$ and $Z$ are conditionally independent given $X$. We assume that the integral (or summing if $X, Y$ or $Z$ are discrete random variables) is on $\mathcal{X} \times \mathcal{Y} \times \mathcal{Z}$. We use $x$, $y$ and $z$ to denote the instances of the respective random variables. The above settings are used throughout the paper. We can view the IB objective $\text{IB}_\beta[p(z|x)]$ (Eq. 1) as a functional of the encoding distribution $p(z|x)$. To prepare for the introduction of IB phase transitions, we first define *relative perturbation function* and *second variation*, as follows.

**Definition 1.** *Relative perturbation function: For $p(z|x)$, its relative perturbation function $r(z|x)$ is a bounded function that maps $\mathcal{X} \times \mathcal{Z}$ to $\mathbb{R}$ and satisfies $\mathbb{E}_{z \sim p(z|x)}[r(z|x)] = 0$. Formally, define $\mathcal{Q}_{\mathcal{Z}|\mathcal{X}} := \{r(z|x) : \mathcal{X} \times \mathcal{Z} \to \mathbb{R} \mid \mathbb{E}_{z \sim p(z|x)}[r(z|x)] = 0, \text{and} \exists M > 0 \text{ s.t. } \forall X \in \mathcal{X}, Z \in \mathcal{Z}, |r(z|x)| \leq M\}$. We have that $r(z|x) \in \mathcal{Q}_{\mathcal{Z}|\mathcal{X}}$ iff $r(z|x)$ is a relative perturbation function of $p(z|x)$. The perturbed probability (density) is $p'(z|x) = p(z|x)(1 + \epsilon \cdot r(z|x))$ for some $\epsilon > 0$.*

**Definition 2.** *Second variation: Let functional $F[f(x)]$ be defined on some normed linear space $\mathcal{R}$. Let us add a perturbative function $\epsilon \cdot h(x)$ to $f(x)$, and now the functional $F[f(x) + \epsilon \cdot h(x)]$ can be expanded as*

$$\Delta F[f(x)] = F[f(x) + \epsilon \cdot h(x)] - F[f(x)]$$
$$= \varphi_1[\epsilon \cdot h(x)] + \varphi_2[\epsilon \cdot h(x)] + \varphi_r[\epsilon \cdot h(x)] \|\epsilon \cdot h(x)\|^2$$

*such that $\lim_{\epsilon \to 0} \varphi_r[\epsilon \cdot h(x)] = 0$, where $\|\cdot\|$ denotes the norm, $\varphi_1[\epsilon \cdot h(x)] = \epsilon \frac{dF[f(x)]}{d\epsilon}$ is a linear functional of $\epsilon \cdot h(x)$, and is called the first variation, denoted as $\delta F[f(x)]$. $\varphi_2[\epsilon \cdot h(x)] = \frac{1}{2}\epsilon^2 \frac{d^2 F[f(x)]}{d\epsilon^2}$ is a quadratic functional of $\epsilon \cdot h(x)$, and is called the second variation, denoted as $\delta^2 F[f(x)]$.*

We can think of the perturbation function $\epsilon \cdot h(x)$ as an infinite-dimensional "vector" ($x$ being the indices), with $\epsilon$ being its amplitude and $h(x)$ its direction. With the above preparations, we define the IB phase transition as a change in the local curvature on the global minimum of $\text{IB}_\beta[p(z|x)]$.

**Definition 3.** *IB phase transitions:* *Let $r(z|x) \in \mathcal{Q}_{\mathcal{Z}|\mathcal{X}}$ be a perturbation function of $p(z|x)$, $p_\beta^*(z|x)$ denote the optimal solution of $\text{IB}_\beta[p(z|x)]$ at $\beta$, where the IB functional $\text{IB}[\cdot]$ is defined in Eq. (1). The IB phase transitions $\beta_i^c$ are the $\beta$ values satisfying the following two conditions:*

*(1) $\forall r(z|x) \in \mathcal{Q}_{\mathcal{Z}|\mathcal{X}}$, $\delta^2 IB_\beta[p(z|x)]\big|_{p_\beta^*(z|x)} \geq 0$;*

*(2) $\lim_{\beta' \to \beta^+} \inf_{r(z|x) \in \mathcal{Q}_{\mathcal{Z}|\mathcal{X}}} \delta^2 IB_{\beta'}[p(z|x)]\big|_{p_\beta^*(z|x)} = 0^-$.*

*Here $\beta^+$ and $0^-$ denote one-sided limits.*

We can understand the $\delta^2 \text{IB}_\beta[p(z|x)]$ as a local "curvature" of the IB objective $\text{IB}_\beta$ (Eq. 1) w.r.t. $p(z|x)$, along some relative perturbation $r(z|x)$. A phase transition occurs when the convexity of $\text{IB}_\beta[p(z|x)]$ w.r.t. $p(z|x)$ changes from a minimum to a saddle point in the neighborhood of its optimal solution $p_\beta^*(z|x)$ as $\beta$ increases from $\beta_c$ to $\beta_c + 0^+$. This means that there exists a perturbation to go downhill and find a better minimum. We validate this definition empirically below.

## 3.2 CONDITION FOR IB PHASE TRANSITIONS

The definition for IB phase transition (Definition 3) indicates the important role $\delta^2 \text{IB}_\beta[p(z|x)]$ plays on the optimal solution in providing the condition for phase transitions. To concretize it and prepare for a more practical condition for IB phase transitions, we expand $\text{IB}_\beta[p(z|x)(1 + \epsilon \cdot r(z|x))]$ to the second order of $\epsilon$, giving:

**Lemma 0.1.** *For $\text{IB}_\beta[p(z|x)]$, the condition of $\forall r(z|x) \in \mathcal{Q}_{\mathcal{Z}|\mathcal{X}}$, $\delta^2 IB_\beta[p(z|x)] \geq 0$ is equivalent to $\beta \leq G[p(z|x)]$. The threshold function $G[p(z|x)]$ is given by:*

$$G[p(z|x)] := \inf_{r(z|x) \in \mathcal{Q}_{\mathcal{Z}|\mathcal{X}}} \mathcal{G}[r(z|x); p(z|x)]$$

$$\mathcal{G}[r(z|x); p(z|x)] := \frac{\mathbb{E}_{x,z \sim p(x,z)}[r^2(z|x)] - \mathbb{E}_{z \sim p(z)}\left[\left(\mathbb{E}_{x \sim p(x|z)}[r(z|x)]\right)^2\right]}{\mathbb{E}_{y,z \sim p(y,z)}\left[\left(\mathbb{E}_{x \sim p(x|y,z)}[r(z|x)]\right)^2\right] - \mathbb{E}_{z \sim p(z)}\left[\left(\mathbb{E}_{x \sim p(x|z)}[r(z|x)]\right)^2\right]} \tag{2}$$

The proof is given in Appendix B, in which we also give Eq. (20) for empirical estimation. Note that Lemma 0.1 is very general and can be applied to any $p(z|x)$, not only at the optimal solution $p_\beta^*(z|x)$.

**The Fisher Information matrix.** In practice, the encoder $p_{\boldsymbol{\theta}}(z|x)$ is usually parameterized by some parameter vector $\boldsymbol{\theta} = (\theta_1, \theta_2, ...\theta_k)^T \in \Theta$, e.g. weights and biases in a neural net, where $\Theta$ is the parameter field. An infinitesimal change of $\boldsymbol{\theta}' \leftarrow \boldsymbol{\theta} + \Delta\boldsymbol{\theta}$ induces a relative perturbation $\epsilon \cdot r(z|x) \simeq \Delta\boldsymbol{\theta}^T \frac{\partial \log p_{\boldsymbol{\theta}}(z|x)}{\partial \boldsymbol{\theta}}$ on $p_{\boldsymbol{\theta}}(z|x)$, from which we can compute the threshold function $G_\Theta[p_{\boldsymbol{\theta}}(z|x)]$:

**Lemma 0.2.** *For $\text{IB}_\beta[p_{\boldsymbol{\theta}}(z|x)]$ objective, the condition of $\forall \Delta\boldsymbol{\theta} \in \Theta$, $\delta^2 IB_\beta[p_{\boldsymbol{\theta}}(z|x)] \geq 0$ is equivalent to $\beta \leq G_\Theta[p_{\boldsymbol{\theta}}(z|x)]$, where*

$$G_\Theta[p_{\boldsymbol{\theta}}(z|x)] := \inf_{\Delta\boldsymbol{\theta} \in \Theta} \frac{\Delta\boldsymbol{\theta}^T \left(\mathcal{I}_{Z|X}(\boldsymbol{\theta}) - \mathcal{I}_Z(\boldsymbol{\theta})\right) \Delta\boldsymbol{\theta}}{\Delta\boldsymbol{\theta}^T \left(\mathcal{I}_{Z|Y}(\boldsymbol{\theta}) - \mathcal{I}_Z(\boldsymbol{\theta})\right) \Delta\boldsymbol{\theta}} = \lambda_{\max}^{-1} \tag{3}$$

*where $\mathcal{I}_Z(\boldsymbol{\theta}) := \int dz p_{\boldsymbol{\theta}}(z) \left(\frac{\partial \log p_{\boldsymbol{\theta}}(z)}{\partial \boldsymbol{\theta}}\right) \left(\frac{\partial \log p_{\boldsymbol{\theta}}(z)}{\partial \boldsymbol{\theta}}\right)^T$ is the Fisher information matrix of $\boldsymbol{\theta}$ for $Z$, $\mathcal{I}_{Z|X}(\boldsymbol{\theta}) := \int dx dz p(x) p_{\boldsymbol{\theta}}(z|x) \left(\frac{\partial \log p_{\boldsymbol{\theta}}(z|x)}{\partial \boldsymbol{\theta}}\right) \left(\frac{\partial \log p_{\boldsymbol{\theta}}(z|x)}{\partial \boldsymbol{\theta}}\right)^T$, $\mathcal{I}_{Z|Y}(\boldsymbol{\theta}) := \int dy dz p(y) p_{\boldsymbol{\theta}}(z|y) \left(\frac{\partial \log p_{\boldsymbol{\theta}}(z|y)}{\partial \boldsymbol{\theta}}\right) \left(\frac{\partial \log p_{\boldsymbol{\theta}}(z|y)}{\partial \boldsymbol{\theta}}\right)^T$ are the conditional Fisher information matrix (Zegers, 2015) of $\boldsymbol{\theta}$ for $Z$ conditioned on $X$ and $Y$, respectively. $\lambda_{\max}$ is the largest eigenvalue of $C^{-1} \left(\mathcal{I}_{Z|Y}(\boldsymbol{\theta}) - \mathcal{I}_Z(\boldsymbol{\theta})\right)(C^T)^{-1}$ with $v_{\max}$ the corresponding eigenvector, where $CC^T$ is the Cholesky decomposition of the matrix $\mathcal{I}_{Z|X}(\boldsymbol{\theta}) - \mathcal{I}_Z(\boldsymbol{\theta})$, and $v_{\max}$ is the eigenvector for $\lambda_{\max}$. The infimum is attained at $\Delta\boldsymbol{\theta} = (C^T)^{-1} v_{\max}$.*

The proof is in appendix C. We see that for parameterized encoders $p_\theta(z|x)$, each term of $G[p(z|x)]$ in Eq. (2) can be replaced by a bilinear form with the Fisher information matrix of the respective variables. Although this lemma is not required to understand the more general setting of Lemma 0.1, where the model is described in a functional space, Lemma 0.2 helps understand $G[p(z|x)]$ for parameterized models, which permits directly linking the phase transitions to the model's parameters.

**Phase Transitions.** Now we introduce Theorem 1 that gives a concrete and practical condition for IB phase transitions, which is the core result of the paper:

**Theorem 1.** *The IB phase transition points $\{\beta_i^c\}$ as defined in Definition 3 are given by the roots of the following equation:*

$$G[p_\beta^*(z|x)] = \beta \tag{4}$$

*where $G[p(z|x)]$ is given by Eq. (2) and $p_\beta^*(z|x)$ is the optimal solution of $IB_\beta[p(z|x)]$ at $\beta$.*

We can understand Eq. (4) as the condition when $\delta^2 IB_\beta[p(z|x)]$ is *about* to be able to be negative at the optimal solution $p_\beta^*(z|x)$ for a given $\beta$. The proof for Theorem 1 is given in Appendix D. In Section 4, we will analyze Theorem 1 in detail.

## 4 UNDERSTANDING THE FORMULA FOR IB PHASE TRANSITIONS

In this section we set out to understand $G[p(z|x)]$ as given by Eq. (2) and the phase transition condition as given by Theorem 1, from the perspectives of Jensen's inequality and representational maximum correlation.

### 4.1 JENSEN'S INEQUALITY

The condition for IB phase transitions given by Theorem 1 involves $G[p(z|x)] = \inf_{r(z|x) \in \mathcal{Q}_{\mathcal{Z}|\mathcal{X}}} \mathcal{G}[r(z|x); p(z|x)]$ which is in itself an optimization problem. We can understand $G[p(z|x)] = \inf_{r(z|x) \in \mathcal{Q}_{\mathcal{Z}|\mathcal{X}}} \frac{A-C}{B-C}$ in Eq. (2) using Jensen's inequality:

$$\underbrace{\mathbb{E}_{x,z \sim p(x,z)}[r^2(z|x)]}_{A} \geq \underbrace{\mathbb{E}_{y,z \sim p(y,z)}\left[\left(\mathbb{E}_{x \sim p(x|y,z)}[r(z|x)]\right)^2\right]}_{B} \geq \underbrace{\mathbb{E}_{z \sim p(z)}\left[\left(\mathbb{E}_{x \sim p(x|z)}[r(z|x)]\right)^2\right]}_{C}$$

$$\tag{5}$$

The equality between $A$ and $B$ holds when the perturbation $r(z|x)$ is constant w.r.t. $x$ for any $z$; the equality between $B$ and $C$ holds when $\mathbb{E}_{x \sim p(x|y,z)}[r(z|x)]$ is constant w.r.t. $y$ for any $z$. Therefore, the minimization of $\frac{A-C}{B-C}$ encourages the relative perturbation function $r(z|x)$ to be as constant w.r.t. $x$ as possible (minimizing intra-class difference), but as different w.r.t. different $y$ as possible (maximizing inter-class difference), resulting in a *clustering* of the values of $r(z|x)$ for different examples $x$ according to their class $y$. Because of this clustering property in classification problems, we conjecture that there are at most $|\mathcal{Y}| - 1$ phase transitions, where $|\mathcal{Y}|$ is the number of classes, with each phase transition differentiating one or more classes.

### 4.2 REPRESENTATIONAL MAXIMUM CORRELATION

Under certain conditions we can further simplify $G[p(z|x)]$ and gain a deeper understanding of it. Firstly, inspired by maximum correlation (Anantharam et al., 2013), we introduce two new concepts, *representational maximum correlation* and *conditional maximum correlation*, as follows.

**Definition 4.** *Given a joint distribution $p(X, Y)$, and a representation $Z$ satisfying the Markov chain $Z - X - Y$, the representational maximum correlation $\rho_r(X, Y; Z)$ is defined as*

$$\rho_r(X, Y; Z) := \sup_{(f(x,z), g(y,z)) \in \mathcal{S}_1} \mathbb{E}_{x,y,z \sim p(x,y,z)}[f(x, z)g(y, z)] \tag{6}$$

*where $\mathcal{S}_1 = \{(f : \mathcal{X} \times \mathcal{Z} \to \mathbb{R}, g : \mathcal{Y} \times \mathcal{Z} \to \mathbb{R}) \,|\, f, g \text{ bounded}, \text{and } \mathbb{E}_{x \sim p(x|z)}[f(x, z)] = \mathbb{E}_{y \sim p(y|z)}[g(y, z)] = 0, \mathbb{E}_{x,z \sim p(x,z)}[f^2(x, z)] = \mathbb{E}_{y,z \sim p(y,z)}[g^2(y, z)] = 1\}$.*

*The conditional maximum correlation $\rho_m(X, Y|Z)$ is defined as:*

$$\rho_m(X, Y|Z) := \sup_{(f(x),g(y))\in\mathcal{S}_2} \mathbb{E}_{x,y\sim p(x,y|z)}[f(x)g(y)] \tag{7}$$

*where $\mathcal{S}_2 = \{(f : \mathcal{X} \to \mathbb{R}, g : \mathcal{Y} \to \mathbb{R}) \,|\, f, g \text{ bounded, and } \forall z \in \mathcal{Z} : \mathbb{E}_{x\sim p(x|z)}[f(x)] = \mathbb{E}_{y\sim p(y|z)}[g(y)] = 0, \mathbb{E}_{x\sim p(x|z)}[f^2(x)] = \mathbb{E}_{y\sim p(y|z)}[g^2(y)] = 1\}$.*

We prove the following Theorem 2, which expresses $G[p(z|x)]$ in terms of representational maximum correlation and related quantities, with proof given in Appendix F.

**Theorem 2.** *Define $\mathcal{Q}_{\mathcal{Z}|\mathcal{X}}^{(0)} := \{r(z|x) : \mathcal{X} \times \mathcal{Z} \to \mathbb{R} \,|\, r \text{ bounded}\}$. If $\mathcal{Q}_{\mathcal{Z}|\mathcal{X}}^{(0)}$ and $\mathcal{Q}_{\mathcal{Z}|\mathcal{X}}$ satisfy:*
*$\forall r(z|x) \in \mathcal{Q}_{\mathcal{Z}|\mathcal{X}}^{(0)}$, there exists[2] $r_1(z|x) \in \mathcal{Q}_{\mathcal{Z}|\mathcal{X}}$, $s(z) \in \{s(z) : \mathcal{Z} \to \mathbb{R} \,|\, s \text{ bounded}\}$ s.t. $r(z|x) = r_1(z|x) + s(z)$, then we have:*

   *(i) The representation maximum correlation and $G$:*

$$G[p(z|x)] = \frac{1}{\rho_r^2(X, Y; Z)} \tag{8}$$

   *(ii) The representational maximum correlation and conditional maximum correlation:*

$$\rho_r(X, Y; Z) = \sup_{Z\in\mathcal{Z}} [\rho_m(X, Y|Z)] \tag{9}$$

   *(iii) When $Z$ is continuous, an optimal relative perturbation function $r(z|x)$ for $G[p(z|x)]$ is given by*

$$r^*(z|x) = h^*(x)\sqrt{\frac{\delta(z - z^*)}{p(z)}} \tag{10}$$

   *where $z^* = \arg\max_{z\in\mathcal{Z}} \rho_m(X, Y|Z = z)$, and $h^*(x)$ is the optimal solution for the learnability threshold function $h^*(x) = \arg\min_{h(x)\in\{h:\mathcal{X}\to\mathbb{R} \,|\, h \text{ bounded}\}} \beta_0[h(x)]$ with $p(X, Y|Z = z^*)$ ($\beta_0[h(x)]$ is given in Theorem 4 of Wu et al. (2019)).*

   *(iv) For discrete $X$, $Y$ and $Z$, we have*

$$\rho_r(X, Y; Z) = \max_{Z\in\mathcal{Z}} \sigma_2(Z) \tag{11}$$

   *where $\sigma_2(Z)$ is the second largest singular value of the matrix $Q_{X,Y|Z} := \left(\frac{p(x,y|z)}{\sqrt{p(x|z)p(y|z)}}\right)_{x,y} = \left(\frac{p(x,y)}{\sqrt{p(x)p(y)}}\sqrt{\frac{p(z|x)}{p(z|y)}}\right)_{x,y}.$*

Theorem 2 furthers our understanding of $G[p(z|x)]$ and the phase transition condition (Theorem 1), which we elaborate as follows.

**Discovering maximum correlation in the orthogonal space of a learned representation:** Intuitively, the representational maximum correlation measures the maximum linear correlation between $f(X, Z)$ and $g(Y, Z)$ among all real-valued functions $f, g$, under the constraint that $f(X, Z)$ is "orthogonal" to $p(X|Z)$ and $g(Y, Z)$ is "orthogonal" to $p(Y|Z)$. Theorem 2 (i) reveals that $G[p(z|x)]$ is the inverse square of this representational maximum correlation. Theorem 2 (ii) further shows that $G[p(z|x)]$ is finding a specific $z^*$ on which maximum (nonlinear) correlation between $X$ and $Y$

---

[2]For discrete $X$, $Z$ such that the cardinality $|\mathcal{Z}| \geq |\mathcal{X}|$, this is generally true since in this scenario, $h(x, z)$ and $s(z)$ have $|\mathcal{X}||\mathcal{Z}| + |\mathcal{Z}|$ unknown variables, but the condition has only $|\mathcal{X}||\mathcal{Z}| + |\mathcal{X}|$ linear equations. The difference between $\mathcal{Q}_{\mathcal{Z}|\mathcal{X}}$ and $\mathcal{Q}_{\mathcal{Z}|\mathcal{X}}^{(0)}$ is that $\mathcal{Q}_{\mathcal{Z}|\mathcal{X}}^{(0)}$ does not have the requirement of $\mathbb{E}_{p(z|x)}[r(z|x)] = 0$. Combined with Lemma 2.2, this condition allows us to replace $r(z|x) \in \mathcal{Q}_{\mathcal{Z}|\mathcal{X}}$ by $r(z|x) \in \mathcal{Q}_{\mathcal{Z}|\mathcal{X}}^{(0)}$ in Eq. (2).

conditioned on $Z$ can be found. Combined with Theorem 1, we have that when we continuously increase $\beta$, for the optimal representation $Z_\beta^*$ given by $p_\beta^*(z|x)$ at $\beta$, $\rho_r(X, Y; Z_\beta^*)$ shall monotonically decrease due to that $X$ and $Y$ has to find their maximum correlation on the orthogonal space of an increasingly better representation $Z_\beta^*$ that captures more information about $X$. A phase transition occurs when $\rho_r(X, Y; Z_\beta^*)$ reduces to $\frac{1}{\sqrt{\beta}}$, after which as $\beta$ continues to increase, $\rho_r(X, Y; Z_\beta^*)$ will try to find maximum correlation between $X$ and $Y$ orthogonal to the full previously learned representation. This is reminiscent of canonical-correlation analysis (CCA) (Hotelling, 1992) in linear settings, where components with decreasing linear maximum correlation that are orthogonal to previous components are found one by one. In comparison, we show that in IB, each phase transition corresponds to learning a new *nonlinear* component of maximum correlation between $X$ and $Y$ in $Z$, orthogonal to the previously-learned $Z$. In the case of classification where different classes may have different difficulty (e.g. due to label noise or support overlap), we should expect that classes that are less difficult as measured by a larger maximum correlation between $X$ and $Y$ are learned earlier.

**Conspicuous subset conditioned on a single $z$:**  Furthermore, we show in (iii) that an optimal relative perturbation function $r(z|x)$ can be decomposed into a product of two factors, a $\sqrt{\frac{\delta(z-z^*)}{p(z)}}$ factor that only focus on perturbing a specific point $z^*$ in the representation space, and an $h^*(x)$ factor that is finding the "conspicuous subset" (Wu et al., 2019), i.e. the most confident, large, typical, and imbalanced subset in the $X$ space for the distribution $(X, Y) \sim p(X, Y|z^*)$.

**Singular values**  In categorical settings, (iv) reveals a connection between $G[p(z|x)]$ and the singular value of the $Q_{X,Y|Z}$ matrix. Due to the property of SVD, we know that the square of the singular values of $Q_{X,Y|Z}$ equals the non-negative eigenvalue of the matrix $Q_{X,Y|Z}^T Q_{X,Y|Z}$. Then the phase transition condition in Theorem 1 is equivalent to a (nonlinear) eigenvalue problem. This is resonant with previous analogy with CCA in linear settings, and is also reminiscent of the linear eigenvalue problem in Gaussian IB (Chechik et al., 2005).

## 5 ALGORITHM FOR PHASE TRANSITIONS DISCOVERY IN CLASSIFICATION

As a consequence of the theoretical analysis above, we are able to derive an algorithm to efficiently estimate the phase transitions for a given model architecture and dataset. This algorithm also permits us to empirically confirm some of our theoretical results in Section 6.

Typically, classification involves high-dimensional inputs $X$. Without sweeping the full range of $\beta$ where at each $\beta$ it is a full learning problem, it is in general a difficult task to estimate the phase transitions. In Algorithm 1, we present a two-stage approach.

In the first stage, we train a single maximum likelihood neural network $f_{\boldsymbol{\theta}}$ with the same encoder architecture as in the (variational) IB to estimate $p(y|x)$, and obtain an $N \times C$ matrix $p(y|x)$, where $N$ is the number of examples in the dataset and $C$ is the number of classes. In the second stage, we perform an iterative algorithm w.r.t. $G$ and $\beta$, alternatively, to converge to a phase transition point.

Specifically, for a given $\beta$, we use a Blahut-Arimoto type IB algorithm (Tishby et al., 2000) to efficiently reach IB optimal $p_\beta^*(z|x)$ at $\beta$, then use SVD (with the formula given in Theorem 2 (iv)) to efficiently estimate $G[p_\beta^*(z|x)]$ at $\beta$ (step 8). We then use the $G[p_\beta^*(z|x)]$ value as the new $\beta$ and do it again (step 7 in the next iteration). At convergence, we will reach the phase transition point given by $G[p_\beta^*(z|x)] = \beta$ (Theorem 1). After convergence as measured by patience parameter $K$, we slightly increase $\beta$ by $\delta$ (step 13), so that the algorithm can discover the subsequent phase transitions.

## 6 EMPIRICAL STUDY

We quantitatively and qualitatively test the ability of our theory and Algorithm 1 to provide good predictions for IB phase transitions. We first verify them in fully categorical settings, where $X, Y, Z$ are all discrete, and we show that the phase transitions can correspond to learning new classes as we increase $\beta$. We then test our algorithm on versions of the MNIST and CIFAR10 datasets with added label noise.

---

**Algorithm 1 Phase transitions discovery for IB**

---

**Require** $(X, Y)$: the dataset
**Require** $f_{\boldsymbol{\theta}}$: a neural net with the same encoder architecture as the (variational) IB
**Require** $K$: patience
**Require** $\delta$: precision floor
**Require** $R$: maximum ratio between $\beta^{(\text{th})}$ and $\beta$.
// *First stage: fit $p(y|x)$ using neural net $f_{\boldsymbol{\theta}}$:*
1: $p(y|x) \leftarrow$ fitting $(X, Y)$ using $f_{\boldsymbol{\theta}}$ via maximum likelihood.
2: $p(x) \leftarrow \frac{1}{N}$

// *Second stage: coordinate descent using $G[p(z|x)]$ and IB algorithm:*
3: $\beta_0^c \leftarrow \beta^{(\text{th})}(1)$
4: $\mathbb{B} \leftarrow \{\beta_0^c\}$   //$\mathbb{B}$ *is a set collecting the phase transition points*
5: $(\beta^{(\text{new})}, \beta, \text{count}) \leftarrow (\beta_0^c, 1, 0)$
6: **while** $\frac{\beta^{(\text{new})}}{\beta} < R$ **do**:
7:      $\beta \leftarrow \beta^{(\text{new})}$
8:      $\beta^{(\text{new})} \leftarrow \beta^{(\text{th})}(\beta)$
9:      **if** $\beta^{(\text{new})} - \beta < \delta$ **do**:
10:          $\text{count} \leftarrow \text{count} + 1$
11:          **if** $\text{count} > K$ **do**:
12:              $\mathbb{B} \leftarrow \mathbb{B} \cup \{\beta^{(\text{new})}\}$
13:              $\beta^{(\text{new})} \leftarrow \beta^{(\text{new})} + \delta$
14:          **end if**
15:      **else**: $\text{count} \leftarrow 0$
16:      **end if**
17: **end while**
18: **return** $\mathbb{B}$

**subroutine** $\beta^{(\text{th})}(\beta)$:
s1: Compute $p_\beta^*(z|x)$ using the IB algorithm (Tishby et al., 2000).
s2: $\beta^{(\text{new})} \leftarrow G[p_\beta^*(z|x)]$ using SVD (Eq. 8 and 11).
s3: **return** $\beta^{(\text{new})}$

---

### 6.1 CATEGORICAL DATASET

For categorical datasets, $X$ and $Y$ are discrete, and $p(X)$ and $p(Y|X)$ are given. To test Theorem 1, we use the Blahut-Arimoto IB algorithm to compute the optimal $p_\beta^*(z|x)$ for each $\beta$. $I(Y; Z^*)$ vs. $\beta$ is plotted in Fig. 2 (a). There are two phase transitions at $\beta_0^c$ and $\beta_1^c$. For each $\beta$ and the corresponding $p_\beta^*(z|x)$, we use the SVD formula (Theorem 2) to compute $G[p_\beta^*(z|x)]$, shown in Fig. 2 (b). We see that $G[p_\beta^*(z|x)] = \beta$ at exactly the observed phase transition points $\beta_0^c$ and $\beta_1^c$. Moreover, starting at $\beta = 1$, Alg. 1 converges to each phase transition points within few iterations. Our other experiments with random categorical datasets show similarly tight matches.

Furthermore, in Appendix G we show that the phase transitions correspond to the onset of separation of $p(z|x)$ for subsets of $X$ that correspond to different classes. This supports our conjecture from Section 4.1 that there are at most $|\mathcal{Y}| - 1$ phase transitions in classification problems.

### 6.2 MNIST DATASET

For continuous $X$, how does our algorithm perform, and will it reveal aspects of the dataset? We first test our algorithm in a 4-class MNIST with noisy labels[3], whose confusion matrix and experimental settings are given in Appendix H. Fig. 3 (a) shows the path Alg. 1 takes. We see again that in each

---

[3]We use 4 classes since it is simpler than the full 10 classes, but still potentially possesses phase transitions. We use noisy label to mimic realistic settings where the data may be noisy and also to have controllable difficulty for different classes.

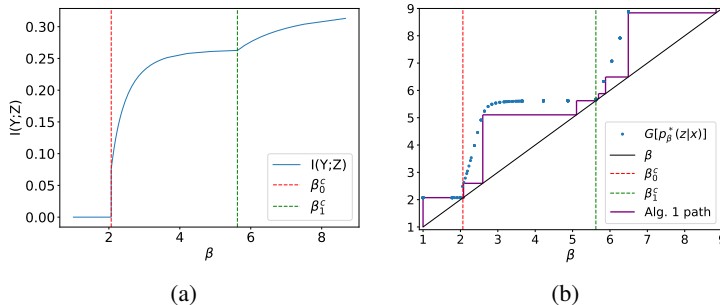

(a)                                                            (b)

Figure 2: (a) $I(Y; Z^*)$ vs. $\beta$ for a categorical dataset with $|X| = |Y| = |Z| = 3$, where $Z^*$ is given by $p_\beta^*(z|x)$, and the vertical lines are the experimentally discovered phase transition points $\beta_0^c$ and $\beta_1^c$. (b) $G[p_\beta^*(z|x)]$ vs. $\beta$ for the same dataset, and the path for Alg. 1, with $\beta_0^c$ and $\beta_1^c$ in (a) also plotted. The dataset is given in Fig. 5.

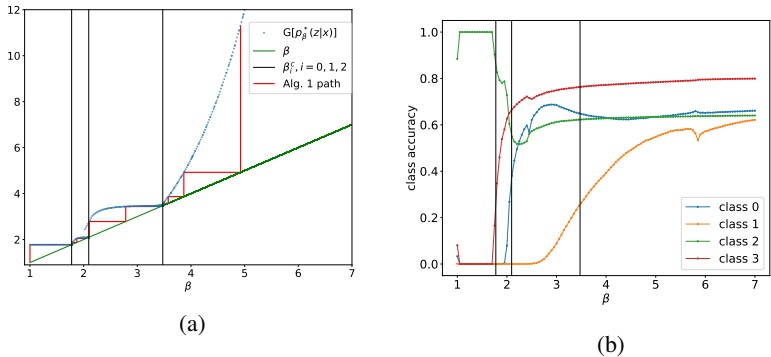

(a)                                                            (b)

Figure 3: (a) Path of Alg. 1 starting with $\beta = 1$, where the maximum likelihood model $f_\theta$ is using the same encoder architecture as in the CEB model. This stairstep path shows that Alg. 1 is able to ignore very large regions of $\beta$, while quickly and precisely finding the phase transition points. Also plotted is an accumulation of $G[p_\beta^*(z|x)]$ vs. $\beta$ by running Alg. 1 with varying starting $\beta$ (blue dots). (b) Per-class accuracy vs. $\beta$, where the accuracy at each $\beta$ is from training an independent CEB model on the dataset. The per-class accuracy denotes the fraction of correctly predicted labels by the CEB model for the observed label $\tilde{y}$.

phase Alg. 1 converges to the phase transition points within a few iterations, and it discovers in total 3 phase transition points. Similar to the categorical case, we expect that each phase transition corresponds to the onset of learning a new class, and that the last class is much harder to learn due to a larger separation of $\beta$. Therefore, this class should have a much larger label noise so that it is hard to capture this component of maximum correlation between $X$ and $Y$, as analyzed in representational maximum correlation (Section 4.2). Fig. 3 (b) plots the per-class accuracy with increasing $\beta$ for running the Conditional Entropy Bottleneck (Fischer, 2018) (another variational bound on IB). We see that the first two predicted phase transition points $\beta_0^c$, $\beta_1^c$ closely match the observed onset of learning class 3 and class 0. Class 1 is observed to learn earlier than expected, possibly due to the gap between the variational IB objective and the true IB objective in continuous settings. By looking at the confusion matrix for the label noise (Fig. 7), we see that the ordering of onset of learning: class 2, 3, 0, 1, corresponds exactly to the decreasing diagonal element $p(\tilde{y} = 1|y = 1)$ (increasing noise) of the classes, and as predicted, class 1 has a much smaller diagonal element $p(\tilde{y} = 1|y = 1)$ than the other three classes, which makes it much more difficult to learn. This ordering of classes by difficulty is what our representational maximum correlation predicts.

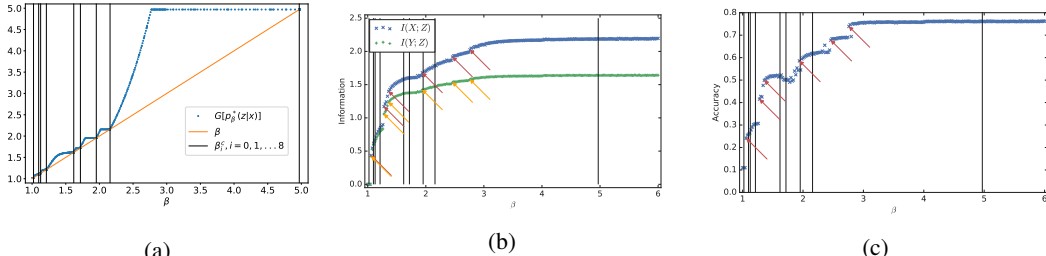

Figure 4: **(a)** Accumulated $G[p_\beta^*(z|x)]$ vs. $\beta$ by running Alg. 1 with varying starting $\beta$ (blue dots). Also plotted are predicted phase transition points. **(b)** $I(X;Z)$ and $I(Y;Z)$ vs. $\beta$. The manually-identified phase transition points are labelled with arrows. The vertical black lines are the phase transitions identified by Alg. 1, denoted as $\beta_0^c$, $\beta_1^c$,... $\beta_8^c$, from left to right. **(c)** Accuracy vs. $\beta$ with the same sets of points identified. The most interesting region is right before $\beta = 2$, where accuracy *decreases* with $\beta$. Alg. 1 identifies both sides of that region, as well as points at or near all of the early obvious phase transitions. It also seems to miss later transitions, possibly due to the gap between the variational IB objective and the true IB objective in continuous settings.

## 6.3    CIFAR10 DATASET

Finally, we investigate the CIFAR10 experiment from Section 1. The details of the experimental setup are described in Appendix I. This experiment stretches the current limits of our discrete approximation to the underlying continuous representation being learned by the models. Nevertheless, we can see in Fig. 4 that many of the visible empirical phase transitions are tightly identified by Alg. 1. Particularly, the onset of learning is predicted quite accurately; the large interval between the predicted $\beta_3 = 1.21$ and $\beta_4 = 1.61$ corresponds well to the continuous increase of $I(X;Z)$ and $I(Y;Z)$ at the same interval. And Alg. 1 is able to identify many dense transitions not obviously seen by just looking at $I(Y;Z)$ vs. $\beta$ curve alone. Alg. 1 predicts 9 phase transitions, exactly equal to $|\mathcal{Y}| - 1$ for CIFAR10.

## 7    CONCLUSION

In this work, we observe and study the phase transitions in IB as we vary $\beta$. We introduce the definition for IB phase transitions, and based on it derive a formula that gives a practical condition for IB phase transitions. We further understand the formula via Jensen's inequality and representational maximum correlation. We reveal the close interplay between the IB objective, the dataset and the learned representation, as each phase transition is learning a nonlinear maximum correlation component in the orthogonal space of the learned representation. We present an algorithm for finding the phase transitions, and show that it gives tight matches with observed phase transitions in categorical datasets, predicts onset of learning new classes and class difficulty in MNIST, and predicts prominent transitions in CIFAR10 experiments. This work is a first theoretical step towards a deeper understanding of the phenomenon of phase transitions in the Information Bottleneck. We believe our approach will be applicable to other "trade-off" objectives, like $\beta$-VAE (Higgins et al., 2017) and InfoDropout (Achille & Soatto, 2018a), where the model's ability to predict is balanced against a measure of complexity.

## 8    ACKNOWLEDGEMENTS

The authors would like to thank Alex Alemi, Kevin Murphy, Sergey Ioffe, Isaac Chuang and Max Tegmark for helpful discussions.

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

# Appendix

## A  CALCULUS OF VARIATIONS AT ANY ORDER OF $\text{IB}_\beta[p(z|x)]$

Here we prove the Lemma 2.1, which will be crucial in the lemmas and theorems in this paper that follows.

**Lemma 2.1.** *For a relative perturbation function $r(z|x) \in \mathcal{Q}_{\mathcal{Z}|\mathcal{X}}$ for a $p(z|x)$, where $r(z|x)$ satisfies $\mathbb{E}_{z \sim p(z|x)}[r(z|x)] = 0$, we have that the IB objective can be expanded as*

$$\text{IB}_\beta[p(z|x)(1 + \epsilon \cdot r(z|x))]$$

$$= \text{IB}_\beta[p(z|x)] + \epsilon \cdot \left( \mathbb{E}_{x,z \sim p(x,z)} \left[ r(z|x) \log \frac{p(z|x)}{p(z)} \right] - \beta \cdot \mathbb{E}_{y,z \sim p(y,z)} \left[ r(z|y) \log \frac{p(z|y)}{p(z)} \right] \right)$$

$$+ \sum_{n=2}^{\infty} \frac{(-1)^n \epsilon^n}{n(n-1)} \left\{ (\mathbb{E}[r^n(z|x)] - \mathbb{E}[r^n(z)]) - \beta \cdot (\mathbb{E}[r^n(z|y)] - \mathbb{E}[r^n(z)]) \right\}$$

$$= \text{IB}_\beta[p(z|x)] + \epsilon \cdot \left( \mathbb{E}_{x,z \sim p(x,z)} \left[ r(z|x) \log \frac{p(z|x)}{p(z)} \right] - \beta \cdot \mathbb{E}_{y,z \sim p(y,z)} \left[ r(z|y) \log \frac{p(z|y)}{p(z)} \right] \right)$$

$$+ \frac{\epsilon^2}{1 \cdot 2} \left\{ \left( \mathbb{E}[r^2(z|x)] - \mathbb{E}[r^2(z)] \right) - \beta \cdot \left( \mathbb{E}[r^2(z|y)] - \mathbb{E}[r^2(z)] \right) \right\}$$

$$- \frac{\epsilon^3}{2 \cdot 3} \left\{ \left( \mathbb{E}[r^3(z|x)] - \mathbb{E}[r^3(z)] \right) - \beta \cdot \left( \mathbb{E}[r^3(z|y)] - \mathbb{E}[r^3(z)] \right) \right\}$$

$$+ \frac{\epsilon^4}{3 \cdot 4} \left\{ \left( \mathbb{E}[r^4(z|x)] - \mathbb{E}[r^4(z)] \right) - \beta \cdot \left( \mathbb{E}[r^4(z|y)] - \mathbb{E}[r^4(z)] \right) \right\}$$

$$- \dots$$

$$(12)$$

*where $r(z|y) = \mathbb{E}_{x \sim p(x|y,z)}[r(z|x)]$ and $r(z) = \mathbb{E}_{x \sim p(x|z)}[r(z|x)]$. The expectations in the equations are all w.r.t. all variables. For example $\mathbb{E}[r^2(z|x)] = \mathbb{E}_{x,z \sim p(x,z)}[r^2(z|x)]$.*

*Proof.* Suppose that we perform a relative perturbation $r(z|x)$ on $p(z|x)$ such that the perturbed conditional probability is $p'(z|x) = p(z|x)(1 + \epsilon \cdot r(z|x))$, then we have

$$p'(z) = \int p(x)p'(z|x)dx = \int dx p(x)p(z|x)(1 + \epsilon \cdot r(z|x)) = p(z) + \epsilon \cdot \int dx p(x)p(z|x)r(z|x)$$

Therefore, we can denote the corresponding relative perturbation $r(z)$ on $p(z)$ as

$$r(z) \equiv \frac{1}{\epsilon} \frac{p'(z) - p(z)}{p(z)} = \frac{1}{p(z)} \int dx p(x)p(z|x)r(z|x) = \mathbb{E}_{x \sim p(x|z)}[r(z|x)]$$

Similarly, we have

$$p'(z|y) = \frac{p'(y,z)}{p(y)} = \frac{1}{p(y)} \int dx p(x,y)p(z|x)(1 + \epsilon \cdot r(z|x)) = p(z|y) + \epsilon \cdot \frac{1}{p(y)} \int dx p(x,y)p(z|x)r(z|x)$$

And we can denote the corresponding relative perturbation $r(z|y)$ on $p(z|y)$ as

$$r(z|y) \equiv \frac{1}{\epsilon} \frac{p'(z|y) - p(z|y)}{p(z|y)} = \frac{1}{p(z|y)p(y)} \int dx p(x,y)p(z|x)r(z|x) = \mathbb{E}_{x \sim p(x|y,z)}[r(z|x)]$$

Since

$$\text{IB}_\beta[p(z|x)] = I(X;Z) - \beta \cdot I(Y;Z) = \int dx dz p(x,z) \log \frac{p(z|x)}{p(z)} - \beta \cdot \int dy dz p(y,z) \log \frac{p(z|y)}{p(z)}$$

We have

$$\mathrm{IB}_\beta[p'(z|x)] = \mathrm{IB}_\beta[p(z|x)(1 + \epsilon \cdot r(z|x))]$$

$$= \int dx dz p(x) p'(z|x) \log \frac{p'(z|x)}{p'(z)} - \beta \cdot \int dy dz p(y) p'(z|y) \log \frac{p'(z|y)}{p'(z)}$$

$$= \int dx dz p(x) p(z|x)(1 + \epsilon \cdot r(z|x)) \log \frac{p(z|x)(1 + \epsilon \cdot r(z|x))}{p(z)(1 + \epsilon \cdot r(z))}$$

$$- \beta \cdot \int dy dz p(y) p(z|y)(1 + \epsilon \cdot r(z|y)) \log \frac{p(z|y)(1 + \epsilon \cdot r(z|y))}{p(z)(1 + \epsilon \cdot r(z))}$$

$$= \int dx dz p(x) p(z|x)(1 + \epsilon \cdot r(z|x)) \left[ \log \frac{p(z|x)}{p(z)} + \log\left(1 + \epsilon \cdot r(z|x)\right) - \log\left(1 + \epsilon \cdot r(z)\right) \right]$$

$$- \beta \cdot \int dy dz p(y) p(z|y)(1 + \epsilon \cdot r(z|y)) \left[ \log \frac{p(z|y)}{p(z)} + \log\left(1 + \epsilon \cdot r(z|y)\right) - \log\left(1 + \epsilon \cdot r(z)\right) \right]$$

$$= \int dx dz p(x) p(z|x)(1 + \epsilon \cdot r(z|x)) \left[ \log \frac{p(z|x)}{p(z)} + \sum_{n=1}^{\infty} (-1)^{n-1} \frac{\epsilon^n}{n} \left( r(z|x) - r(z) \right) \right]$$

$$- \beta \cdot \int dy dz p(y) p(z|y)(1 + \epsilon \cdot r(z|y)) \left[ \log \frac{p(z|y)}{p(z)} + \sum_{n=1}^{\infty} (-1)^{n-1} \frac{\epsilon^n}{n} \left( r(z|y) - r(z) \right) \right]$$

The $0^{\text{th}}$-order term is simply $\mathrm{IB}_\beta[p(z|x)]$. The first order term is

$$\delta \mathrm{IB}_\beta[p(z|x)] = \epsilon \cdot \left( \mathbb{E}_{x,z\sim p(x,z)} \left[ r(z|x) \log \frac{p(z|x)}{p(z)} \right] - \beta \cdot \mathbb{E}_{y,z\sim p(y,z)} \left[ r(z|y) \log \frac{p(z|y)}{p(z)} \right] \right)$$

The $n^{\text{th}}$-order term for $n \geq 2$ is

$$\delta^n \mathrm{IB}_\beta[p(z|x)]$$

$$= (-1)^n \epsilon^n \int dx dz p(x) p(z|x) \left( -\frac{1}{n} \left[ r^n(z|x) - r^n(z) \right] + r(z|x) \frac{1}{n-1} \left[ r^{n-1}(z|x) - r^n(z) \right] \right)$$

$$- \beta \cdot (-1)^n \epsilon^n \int dy dz p(y) p(z|y) \left( -\frac{1}{n} \left[ r^n(z|y) - r^n(z) \right] + r(z|y) \frac{1}{n-1} \left[ r^{n-1}(z|y) - r^n(z) \right] \right)$$

$$= \frac{(-1)^n \epsilon^n}{n(n-1)} \left( \mathbb{E}_{x,z\sim p(x,z)}[r^n(z|x)] - n\mathbb{E}_{x,z\sim p(x,z)}[r(z|x)r^{n-1}(z)] - (n-1)\mathbb{E}_{z\sim p(z)}[r^n(z)] \right)$$

$$- \beta \cdot \frac{(-1)^n \epsilon^n}{n(n-1)} \left( \mathbb{E}_{y,z\sim p(y,z)}[r^n(z|y)] - n\mathbb{E}_{y,z\sim p(y,z)}[r(z|y)r^{n-1}(z)] - (n-1)\mathbb{E}_{z\sim p(z)}[r^n(z)] \right)$$

$$= \frac{(-1)^n \epsilon^n}{n(n-1)} \left\{ (\mathbb{E}[r^n(z|x)] - \mathbb{E}[r^n(z)]) - \beta \cdot (\mathbb{E}[r^n(z|y)] - \mathbb{E}[r^n(z)]) \right\}$$

In the last equality we have used

$$\mathbb{E}_{x,z\sim p(x,z)}[r(z|x)r^{n-1}(z)] = \mathbb{E}_{z\sim p(z)}[r^{n-1}(z)\mathbb{E}_{x\sim p(x|z)}[r(z|x)]] = \mathbb{E}_{z\sim p(z)}[r^{n-1}(z)r(z)] = \mathbb{E}_{z\sim p(z)}[r^n(z)]$$

Combining the terms with all orders, we have

$$\mathrm{IB}_\beta[p(z|x)(1 + \epsilon \cdot r(z|x))]$$

$$= \mathrm{IB}_\beta[p(z|x)] + \epsilon \cdot \left( \mathbb{E}_{x,z\sim p(x,z)} \left[ r(z|x) \log \frac{p(z|x)}{p(z)} \right] - \beta \cdot \mathbb{E}_{y,z\sim p(y,z)} \left[ r(z|y) \log \frac{p(z|y)}{p(z)} \right] \right)$$

$$+ \sum_{n=2}^{\infty} \frac{(-1)^n \epsilon^n}{n(n-1)} \left\{ (\mathbb{E}[r^n(z|x)] - \mathbb{E}[r^n(z)]) - \beta \cdot (\mathbb{E}[r^n(z|y)] - \mathbb{E}[r^n(z)]) \right\}$$

$$\square$$

As a side note, the KL-divergence between $p'(z|x) = p(z|x)(1 + \epsilon \cdot r(z|x))$ and $p(z|x)$ is

$$
\begin{aligned}
\text{KL}\left(p'(z|x)||p(z|x)\right) &= \int dz p(z|x)(1 + \epsilon \cdot r(z|x)) \log \frac{p(z|x)(1 + \epsilon \cdot r(z|x))}{p(z|x)} \\
&= \int dz p(z|x)(1 + \epsilon \cdot r(z|x)) \left( \epsilon \cdot r(z|x) - \frac{\epsilon^2}{2} \cdot r^2(z|x)) + O(\epsilon^3) \right) \\
&= \epsilon \cdot \int dz p(z|x) r(z|x) + \frac{\epsilon^2}{2} \int dz p(z|x) r^2(z|x) + O(\epsilon^3) \\
&= \frac{\epsilon^2}{2} \mathbb{E}_{z \sim p(z|x)}[r^2(z|x)] + O(\epsilon^3)
\end{aligned}
$$

Therefore, to the second order, we have

$$
\mathbb{E}_{x \sim p(x)} \left[ \text{KL}\left(p'(z|x)||p(z|x)\right) \right] = \frac{\epsilon^2}{2} \mathbb{E}[r^2(z|x)] \tag{13}
$$

Similarly, we have $\mathbb{E}_{x \sim p(x)} \left[ \text{KL}\left(p(z|x)||p'(z|x)\right) \right] = \frac{\epsilon^2}{2} \mathbb{E}[r^2(z|x)]$ up to the second order. Using similar procedure, we have up to the second-order,

$$
\mathbb{E}_{y \sim p(y)} \left[ \text{KL}\left(p'(z|y)||p(z|y)\right) \right] = \mathbb{E}_{y \sim p(y)} \left[ \text{KL}\left(p(z|y)||p'(z|y)\right) \right] = \frac{\epsilon^2}{2} \mathbb{E}[r^2(z|y)]
$$

$$
\text{KL}\left(p'(z)||p(z)\right) = \text{KL}\left(p(z)||p'(z)\right) = \frac{\epsilon^2}{2} \mathbb{E}[r^2(z)]
$$

## B   Proof of Lemma 0.1

*Proof.* From Lemma 2.1, we have

$$
\delta^2 \text{IB}_\beta[p(z|x)] = \frac{\epsilon^2}{2} \left\{ \left( \mathbb{E}[r^2(z|x)] - \mathbb{E}[r^2(z)] \right) - \beta \cdot \left( \mathbb{E}[r^2(z|y)] - \mathbb{E}[r^2(z)] \right) \right\} \tag{14}
$$

The condition of

$$
\forall r(z|x) \in \mathcal{Q}_{\mathcal{Z}|\mathcal{X}}, \delta^2 \text{IB}_\beta[p(z|x)] \geq 0 \tag{15}
$$

is equivalent to

$$
\forall r(z|x) \in \mathcal{Q}_{\mathcal{Z}|\mathcal{X}}, \beta \cdot \left( \mathbb{E}[r^2(z|y)] - \mathbb{E}[r^2(z)] \right) \leq \mathbb{E}[r^2(z|x)] - \mathbb{E}[r^2(z)] \tag{16}
$$

Using Jensen's inequality and the convexity of the square function, we have

$$
\begin{aligned}
\mathbb{E}[r^2(z|y)] &= \mathbb{E}_{y,z \sim p(y,z)} \left[ \left( \mathbb{E}_{x \sim p(x|y,z)}[r(z|x)] \right)^2 \right] \\
&= \mathbb{E}_{z \sim p(z)} \left[ \mathbb{E}_{y \sim p(y|z)} \left[ \left( \mathbb{E}_{x \sim p(x|y,z)}[r(z|x)] \right)^2 \right] \right] \\
&\geq \mathbb{E}_{z \sim p(z)} \left[ \left( \mathbb{E}_{y \sim p(y|z)} \left[ \mathbb{E}_{x \sim p(x|y,z)}[r(z|x)] \right] \right)^2 \right] \\
&= \mathbb{E}_{z \sim p(z)} \left[ \left( \mathbb{E}_{x \sim p(x|z)}[r(z|x)] \right)^2 \right] \\
&= \mathbb{E}[r^2(z)]
\end{aligned}
$$

The equality holds iff $r(z|y) = \mathbb{E}_{x \sim p(x|y,z)}[r(z|x)]$ is constant w.r.t. $y$, for any $z$.

Using Jensen's inequality on $\mathbb{E}[r^2(z)]$, we have $\mathbb{E}[r^2(z)] = \mathbb{E}_{z \sim p(z)} \left[ \left( \mathbb{E}_{x \sim p(x|z)}[r(z|x)] \right)^2 \right] \leq \mathbb{E}_{z \sim p(z)} \left[ \mathbb{E}_{x \sim p(x|z)}[r^2(z|x)] \right] = \mathbb{E}[r^2(z|x)]$, where the equality holds iff $r(z|x)$ is constant w.r.t. $x$ for any $z$.

When $\mathbb{E}[r^2(z|y)] - \mathbb{E}[r^2(z)] > 0$, we have that the condition Eq. (16) is equivalent to $\forall r(z|x) \in \mathcal{Q}_{\mathcal{Z}|\mathcal{X}}, \beta \le \frac{\mathbb{E}[r^2(z|x)] - \mathbb{E}[r^2(z)]}{\mathbb{E}[r^2(z|y)] - \mathbb{E}[r^2(z)]}$, i.e.

$$\beta \le G[p(z|x)] \equiv \inf_{r(z|x) \in \mathcal{Q}_{\mathcal{Z}|\mathcal{X}}} \frac{\mathbb{E}[r^2(z|x)] - \mathbb{E}[r^2(z)]}{\mathbb{E}[r^2(z|y)] - \mathbb{E}[r^2(z)]} \tag{17}$$

where $r(z|y) = \mathbb{E}_{x \sim p(x|y,z)}[r(z|x)]$ and $r(z) = \mathbb{E}_{x \sim p(x|z)}[r(z|x)]$.

If $\mathbb{E}[r^2(z|y)] - \mathbb{E}[r^2(z)] = 0$, substituting into Eq. (16), we have

$$\beta \cdot 0 \le \mathbb{E}[r^2(z|x)] - \mathbb{E}[r^2(z)] \tag{18}$$

which is always true due to that $\mathbb{E}[r^2(z|x)] \ge \mathbb{E}[r^2(z)]$, and will be a looser condition than Eq. (17) above. Above all, we have Eq. (17).

$\square$

**Empirical estimate of $G[p(z|x)]$** To empirically estimate $G[p(z|x)]$ from a minibatch of $\{(x_i, y_i)\}, i = 1, 2, \ldots N$ and the encoder $p(z|x)$, we can make the following Monte Carlo importance sampling estimation, where we use the samples $\{x_j\} \sim p(x)$ and also get samples of $\{z_i\} \sim p(z) = p(x)p(z|x)$, and have:

$$\mathbb{E}_{x,z \sim p(x,z)}[r^2(z|x)] = \int dx dz p(x) p(z) \frac{p(x,z)}{p(x)p(z)} r^2(z|x)$$

$$\simeq \frac{1}{N^2} \sum_{i=1}^{N} \sum_{j=1}^{N} \frac{p(x_j, z_i)}{p(x_j)p(z_i)} r^2(z_i|x_j)$$

$$\mathbb{E}_{z \sim p(z)}[r^2(z)] = \mathbb{E}_{z \sim p(z)} \left[ \left( \mathbb{E}_{x \sim p(x|z)}[r(z|x)] \right)^2 \right]$$

$$\simeq \frac{1}{N} \sum_{i=1}^{N} \left( \int dx p(x|z_i) r(z_i|x) \right)^2$$

$$= \frac{1}{N} \sum_{i=1}^{N} \left( \int dx p(x) \frac{p(z_i|x)}{p(z_i)} r(z_i|x) \right)^2$$

$$\simeq \frac{1}{N} \sum_{i=1}^{N} \left( \frac{1}{N} \sum_{j=1}^{N} \frac{p(z_i|x_j)}{p(z_i)} r(z_i|x_j) \right)^2$$

$$\simeq \frac{1}{N} \sum_{i=1}^{N} \left( \frac{1}{N} \sum_{j=1}^{N} \frac{p(z_i|x_j)}{\frac{1}{N} \sum_{k=1}^{N} p(z_i|x_k)} r(z_i|x_j) \right)^2$$

$$= \frac{1}{N} \sum_{i=1}^{N} \left( \frac{\sum_{j=1}^{N} p(z_i|x_j) r(z_i|x_j)}{\sum_{j=1}^{N} p(z_i|x_j)} \right)^2$$

$$\mathbb{E}_{y,z\sim p(y,z)}[r^2(z|y)] = \mathbb{E}_{y,z\sim p(y,z)}\left[\left(\mathbb{E}_{x\sim p(x|y,z)}[r(z|x)]\right)^2\right]$$

$$\simeq \frac{1}{N}\sum_{i=1}^{N}\left(\int dx p(x|y_i,z_i)r(z_i|x)\right)^2$$

$$= \frac{1}{N}\sum_{i=1}^{N}\left(\frac{1}{p(y_i,z_i)}\int dx p(y_i)p(x|y_i)p(z_i|x)r(z_i|x)\right)^2$$

$$= \frac{1}{N}\sum_{i=1}^{N}\left(\frac{\int dx p(y_i)p(x|y_i)p(z_i|x)r(z_i|x)}{\int dx p(y_i)p(x|y_i)p(z_i|x)}\right)^2$$

$$\simeq \frac{1}{N}\sum_{i=1}^{N}\left(\frac{\sum_{x_j\in\Omega_x(y_i)}p(z_i|x_j)r(z_i|x_j)}{\sum_{x_j\in\Omega_x(y_i)}p(z_i|x_j)}\right)^2$$

$$= \frac{1}{N}\sum_{i=1}^{N}\left(\frac{\sum_{j=1}^{N}p(z_i|x_j)r(z_i|x_j)\mathbb{1}\,[y_i=y_j]}{\sum_{j=1}^{N}p(z_i|x_j)\mathbb{1}\,[y_i=y_j]}\right)^2$$

Here $\Omega_x(y_i)$ denotes the set of $x$ examples that has label of $y_i$, and $\mathbb{1}[\cdot]$ is an indicator function that takes value 1 if its argument is true, 0 otherwise.

The requirement of $\mathbb{E}_{z\sim p(z|x)}[r(z|x)] = 0$ yields

$$0 = \mathbb{E}_{z\sim p(z|x)}[r(z|x)] = \int dz p(z)\frac{p(z|x)}{p(z)}r(z|x) \simeq \frac{1}{N}\sum_{i=1}^{N}\frac{p(z_i|x_j)}{p(z_i)}r(z_i|x_j) \qquad (19)$$

for any $x_j$.

Combining all terms, we have that the empirical $\hat{G}[p(z|x)]$ is given by

$$\hat{G}[p(z|x)] = \inf_{r(z|x)\in\mathcal{Q}_{\mathcal{Z}|\mathcal{X}}} \frac{\frac{1}{N}\sum_{i=1}^{N}\sum_{j=1}^{N}\frac{p(x_j,z_i)}{p(x_j)p(z_i)}r^2(z_i|x_j) - \sum_{i=1}^{N}\left(\frac{\sum_{j=1}^{N}p(z_i|x_j)r(z_i|x_j)}{\sum_{j=1}^{N}p(z_i|x_j)}\right)^2}{\sum_{i=1}^{N}\left(\frac{\sum_{j=1}^{N}p(z_i|x_j)r(z_i|x_j)\mathbb{1}[y_i=y_j]}{\sum_{j=1}^{N}p(z_i|x_j)\mathbb{1}[y_i=y_j]}\right)^2 - \sum_{i=1}^{N}\left(\frac{\sum_{j=1}^{N}p(z_i|x_j)r(z_i|x_j)}{\sum_{j=1}^{N}p(z_i|x_j)}\right)^2}$$
$$(20)$$

where $\{z_i\}\sim p(z)$ and $\{x_i\}\sim p(x)$. It is also possible to use different distributions for importance sampling, which will results in different formulas for empirical estimation of $G[p(z|x)]$.

## C $\quad G_\Theta[p_{\boldsymbol{\theta}}(z|x)]$ FOR PARAMETERIZED DISTRIBUTION $p_{\boldsymbol{\theta}}(z|x)$

*Proof.* For the parameterized[4] $p_{\boldsymbol{\theta}}(z|x)$ with $\boldsymbol{\theta}\in\Theta$, after $\boldsymbol{\theta}'\leftarrow\boldsymbol{\theta}+\Delta\boldsymbol{\theta}$, where[5] $\Delta\boldsymbol{\theta}\in\Theta$ is an infinitesimal perturbation on $\boldsymbol{\theta}$, we have that the distribution changes from $p_{\boldsymbol{\theta}}(z|x)$ to $p_{\boldsymbol{\theta}+\Delta\boldsymbol{\theta}}(z|x)$,

---

[4]In this paper, $\boldsymbol{\theta} = (\theta_1,\theta_2,...\theta_k)^T$ and $\frac{\partial p_{\boldsymbol{\theta}}(z|x)}{\partial\boldsymbol{\theta}} = \left(\frac{\partial p_{\boldsymbol{\theta}}(z|x)}{\partial\theta_1}, \frac{\partial p_{\boldsymbol{\theta}}(z|x)}{\partial\theta_2}, ... \frac{\partial p_{\boldsymbol{\theta}}(z|x)}{\partial\theta_k}\right)^T$ are all column vectors. $\frac{\partial^2 p_{\boldsymbol{\theta}}(z|x)}{\partial\boldsymbol{\theta}^2}$ is a $k\times k$ matrix with $(i,j)$ element of $\frac{\partial^2 p_{\boldsymbol{\theta}}(z|x)}{\partial\theta_i\partial\theta_j}$.

[5]Note that since $\Theta$ is a field, it is closed under subtraction, we have $\Delta\boldsymbol{\theta}\in\Theta$.

and thus the relative perturbation on $p_{\boldsymbol{\theta}}(z|x)$ is

$$\epsilon \cdot r(z|x) = \frac{p_{\boldsymbol{\theta}+\Delta\boldsymbol{\theta}}(z|x) - p_{\boldsymbol{\theta}}(z|x)}{p_{\boldsymbol{\theta}}(z|x)}$$

$$= \frac{1}{p_{\boldsymbol{\theta}}(z|x)} \left( p_{\boldsymbol{\theta}}(z|x) + \Delta\boldsymbol{\theta}^T \frac{\partial p_{\boldsymbol{\theta}}(z|x)}{\partial\boldsymbol{\theta}} + \frac{1}{2}\Delta\boldsymbol{\theta}^T \frac{\partial^2 p_{\boldsymbol{\theta}}(z|x)}{\partial\boldsymbol{\theta}^2}\Delta\boldsymbol{\theta} + O(\|\Delta\boldsymbol{\theta}\|^3) - p_{\boldsymbol{\theta}}(z|x) \right)$$

$$\simeq \Delta\boldsymbol{\theta}^T \frac{\partial}{\partial\boldsymbol{\theta}}\log p_{\boldsymbol{\theta}}(z|x) + \frac{1}{2}\Delta\boldsymbol{\theta}^T \frac{1}{p_{\boldsymbol{\theta}}(z|x)}\frac{\partial^2 p_{\boldsymbol{\theta}}(z|x)}{\partial\boldsymbol{\theta}^2}\Delta\boldsymbol{\theta} + O(\|\Delta\boldsymbol{\theta}\|^3)$$

where $\|\Delta\boldsymbol{\theta}\|$ is the norm of $\Delta\boldsymbol{\theta}$ in the parameter field $\Theta$.

Similarly, we have

$$\epsilon \cdot r(z|y) = \Delta\boldsymbol{\theta}^T \frac{\partial}{\partial\boldsymbol{\theta}}\log p_{\boldsymbol{\theta}}(z|y) + \frac{1}{2}\Delta\boldsymbol{\theta}^T \frac{1}{p_{\boldsymbol{\theta}}(z|y)}\frac{\partial^2 p_{\boldsymbol{\theta}}(z|y)}{\partial\boldsymbol{\theta}^2}\Delta\boldsymbol{\theta} + O(\|\Delta\boldsymbol{\theta}\|^3)$$

$$\epsilon \cdot r(z) = \Delta\boldsymbol{\theta}^T \frac{\partial}{\partial\boldsymbol{\theta}}\log p_{\boldsymbol{\theta}}(z) + \frac{1}{2}\Delta\boldsymbol{\theta}^T \frac{1}{p_{\boldsymbol{\theta}}(z)}\frac{\partial^2 p_{\boldsymbol{\theta}}(z)}{\partial\boldsymbol{\theta}^2}\Delta\boldsymbol{\theta} + O(\|\Delta\boldsymbol{\theta}\|^3)$$

Substituting the above expressions into the expansion of $\text{IB}_\beta[p(z|x)]$ in Eq. (12), and preserving to the second order $\|\Delta\boldsymbol{\theta}\|^2$, we have

$$\text{IB}_\beta[p_{\boldsymbol{\theta}}(z|x)(1 + \epsilon \cdot r(z|x))]$$

$$= \text{IB}_\beta[p_{\boldsymbol{\theta}}(z|x)] + \epsilon \cdot \left( \mathbb{E}_{x,z\sim p_{\boldsymbol{\theta}}(x,z)}\left[ r(z|x)\log\frac{p_{\boldsymbol{\theta}}(z|x)}{p_{\boldsymbol{\theta}}(z)} \right] - \beta \cdot \mathbb{E}_{y,z\sim p_{\boldsymbol{\theta}}(y,z)}\left[ r(z|y)\log\frac{p_{\boldsymbol{\theta}}(z|y)}{p_{\boldsymbol{\theta}}(z)} \right] \right)$$

$$+ \frac{\epsilon^2}{1\cdot 2} \left\{ \left( \mathbb{E}_{x,z\sim p_{\boldsymbol{\theta}}(x,z)}[r^2(z|x)] - \mathbb{E}_{z\sim p_{\boldsymbol{\theta}}(z)}[r^2(z)] \right) - \beta \cdot \left( \mathbb{E}_{y,z\sim p_{\boldsymbol{\theta}}(y,z)}[r^2(z|y)] - \mathbb{E}_{z\sim p_{\boldsymbol{\theta}}(z)}[r^2(z)] \right) \right\}$$

$$= \text{IB}_\beta[p_{\boldsymbol{\theta}}(z|x)] + \mathbb{E}_{x,z\sim p_{\boldsymbol{\theta}}(x,z)}\left[ \left( \Delta\boldsymbol{\theta}^T \frac{\partial}{\partial\boldsymbol{\theta}}\log p_{\boldsymbol{\theta}}(z|x) + \frac{1}{2}\Delta\boldsymbol{\theta}^T \frac{1}{p_{\boldsymbol{\theta}}(z|x)}\frac{\partial^2 p_{\boldsymbol{\theta}}(z|x)}{\partial\boldsymbol{\theta}^2}\Delta\boldsymbol{\theta} \right) \log\frac{p_{\boldsymbol{\theta}}(z|x)}{p_{\boldsymbol{\theta}}(z)} \right]$$

$$- \beta \cdot \mathbb{E}_{y,z\sim p_{\boldsymbol{\theta}}(y,z)}\left[ \left( \Delta\boldsymbol{\theta}^T \frac{\partial}{\partial\boldsymbol{\theta}}\log p_{\boldsymbol{\theta}}(z|y) + \frac{1}{2}\Delta\boldsymbol{\theta}^T \frac{1}{p_{\boldsymbol{\theta}}(z|y)}\frac{\partial^2 p_{\boldsymbol{\theta}}(z|y)}{\partial\boldsymbol{\theta}^2}\Delta\boldsymbol{\theta} \right) \log\frac{p_{\boldsymbol{\theta}}(z|y)}{p_{\boldsymbol{\theta}}(z)} \right]$$

$$+ \frac{1}{2}\left( \mathbb{E}_{x,z\sim p_{\boldsymbol{\theta}}(x,z)}\left[ \left( \Delta\boldsymbol{\theta}^T \frac{\partial}{\partial\boldsymbol{\theta}}\log p_{\boldsymbol{\theta}}(z|x) \right)^2 \right] - \mathbb{E}_{z\sim p_{\boldsymbol{\theta}}(z)}\left[ \left( \Delta\boldsymbol{\theta}^T \frac{\partial}{\partial\boldsymbol{\theta}}\log p_{\boldsymbol{\theta}}(z) \right)^2 \right] \right)$$

$$- \frac{\beta}{2}\left( \mathbb{E}_{y,z\sim p_{\boldsymbol{\theta}}(y,z)}\left[ \left( \Delta\boldsymbol{\theta}^T \frac{\partial}{\partial\boldsymbol{\theta}}\log p_{\boldsymbol{\theta}}(z|y) \right)^2 \right] - \mathbb{E}_{z\sim p_{\boldsymbol{\theta}}(z)}\left[ \left( \Delta\boldsymbol{\theta}^T \frac{\partial}{\partial\boldsymbol{\theta}}\log p_{\boldsymbol{\theta}}(z) \right)^2 \right] \right)$$

$$= \text{IB}_\beta[p_{\boldsymbol{\theta}}(z|x)] + \Delta\boldsymbol{\theta}^T \left\{ \mathbb{E}_{x,z\sim p_{\boldsymbol{\theta}}(x,z)}\left[ \log\frac{p_{\boldsymbol{\theta}}(z|x)}{p_{\boldsymbol{\theta}}(z)}\frac{\partial}{\partial\boldsymbol{\theta}}\log p_{\boldsymbol{\theta}}(z|x) \right] - \beta \cdot \mathbb{E}_{x,z\sim p_{\boldsymbol{\theta}}(x,z)}\left[ \log\frac{p_{\boldsymbol{\theta}}(z|x)}{p_{\boldsymbol{\theta}}(z)}\frac{\partial}{\partial\boldsymbol{\theta}}\log p_{\boldsymbol{\theta}}(z|x) \right] \right\}$$

$$+ \frac{1}{2}\Delta\boldsymbol{\theta}^T \left\{ \left( \mathcal{I}_{Z|X}(\boldsymbol{\theta}) - \mathcal{I}_Z(\boldsymbol{\theta}) \right) - \beta\left( \mathcal{I}_{Z|X}(\boldsymbol{\theta}) - \mathcal{I}_Z(\boldsymbol{\theta}) \right) \right\} \Delta\boldsymbol{\theta}$$

In the last equality we have used $\mathbb{E}_{x,z\sim p_{\boldsymbol{\theta}}(x,z)}[\frac{1}{p_{\boldsymbol{\theta}}(z|x)}\frac{\partial^2 p_{\boldsymbol{\theta}}(z|x)}{\partial\boldsymbol{\theta}^2}] = \int dx p(x)\frac{\partial^2}{\partial\boldsymbol{\theta}^2}\int dz p_{\boldsymbol{\theta}}(z|x) = \int dx p(x)\frac{\partial^2}{\partial\boldsymbol{\theta}^2}1 = \mathbf{0}$, and similarly $\mathbb{E}_{y,z\sim p_{\boldsymbol{\theta}}(y,z)}[\frac{1}{p_{\boldsymbol{\theta}}(z|y)}\frac{\partial^2 p_{\boldsymbol{\theta}}(z|y)}{\partial\boldsymbol{\theta}^2}] = \mathbf{0}$. In other words, the $\|\Delta\boldsymbol{\theta}\|^2$ terms in the first-order variation $\delta\text{IB}_\beta[p_{\boldsymbol{\theta}}(z|x)]$ vanish, and the remaining $\|\Delta\boldsymbol{\theta}\|^2$ are all in $\delta^2\text{IB}_\beta[p_{\boldsymbol{\theta}}(z|x)]$. Also in the last expression, $\mathcal{I}_Z(\boldsymbol{\theta}) \equiv \int dz p_{\boldsymbol{\theta}}(z)\left(\frac{\partial\log p_{\boldsymbol{\theta}}(z)}{\partial\boldsymbol{\theta}}\right)\left(\frac{\partial\log p_{\boldsymbol{\theta}}(z)}{\partial\boldsymbol{\theta}}\right)^T$ is the Fisher information matrix of $\boldsymbol{\theta}$ for $Z$, $\mathcal{I}_{Z|X}(\boldsymbol{\theta}) \equiv \int dxdz p(x)p_{\boldsymbol{\theta}}(z|x)\left(\frac{\partial\log p_{\boldsymbol{\theta}}(z|x)}{\partial\boldsymbol{\theta}}\right)\left(\frac{\partial\log p_{\boldsymbol{\theta}}(z|x)}{\partial\boldsymbol{\theta}}\right)^T$, $\mathcal{I}_{Z|Y}(\boldsymbol{\theta}) \equiv \int dydz p(y)p_{\boldsymbol{\theta}}(z|y)\left(\frac{\partial\log p_{\boldsymbol{\theta}}(z|y)}{\partial\boldsymbol{\theta}}\right)\left(\frac{\partial\log p_{\boldsymbol{\theta}}(z|y)}{\partial\boldsymbol{\theta}}\right)^T$ are the conditional Fisher information matrix (Zegers, 2015) of $\boldsymbol{\theta}$ for $Z$ conditioned on $X$ and $Y$, respectively.

Let us look at

$$\delta^2 \mathrm{IB}_\beta[p_{\boldsymbol{\theta}}(z|x)] = \frac{1}{2}\Delta\boldsymbol{\theta}^T \left\{ \left(\mathcal{I}_{Z|X}(\boldsymbol{\theta}) - \mathcal{I}_Z(\boldsymbol{\theta})\right) - \beta\left(\mathcal{I}_{Z|X}(\boldsymbol{\theta}) - \mathcal{I}_Z(\boldsymbol{\theta})\right) \right\} \Delta\boldsymbol{\theta} \tag{21}$$

Firstly, note that $\delta^2 \mathrm{IB}_\beta[p_{\boldsymbol{\theta}}(z|x)]$ is a quadratic function of $\Delta\boldsymbol{\theta}$, and the scale of $\Delta\boldsymbol{\theta}$ does not change the sign of $\delta^2 \mathrm{IB}_\beta[p_{\boldsymbol{\theta}}(z|x)]$, so the condition of $\forall \Delta\boldsymbol{\theta} \in \Theta$, $\delta^2 \mathrm{IB}_\beta[p_{\boldsymbol{\theta}}(z|x)] \geq 0$ is invariant to the scale of $\Delta\boldsymbol{\theta}$, and is describing the "curvature" in the infinitesimal neighborhood of $\boldsymbol{\theta}$. Therefore, $\Delta\boldsymbol{\theta}$ can explore any value in $\Theta$. Secondly, we see that Eq. (21) is a special case of Eq. (14) with $\epsilon \cdot r(z|x) = \Delta\boldsymbol{\theta}^T \frac{\partial}{\partial\boldsymbol{\theta}}\log p_{\boldsymbol{\theta}}(z|x)$. Therefore, The inequalities due to Jensen still hold: $\epsilon^2\left(\mathbb{E}[r^2(z|x)] - \mathbb{E}[r^2(z)]\right) = \Delta\boldsymbol{\theta}^T\left(\mathcal{I}_{Z|X}(\boldsymbol{\theta}) - \mathcal{I}_Z(\boldsymbol{\theta})\right)\Delta\boldsymbol{\theta} \geq 0$, $\epsilon^2\left(\mathbb{E}[r^2(z|y)] - \mathbb{E}[r^2(z)]\right) = \Delta\boldsymbol{\theta}^T\left(\mathcal{I}_{Z|Y}(\boldsymbol{\theta}) - \mathcal{I}_Z(\boldsymbol{\theta})\right)\Delta\boldsymbol{\theta} \geq 0$. If $\Delta\boldsymbol{\theta}^T\left(\mathcal{I}_{Z|Y}(\boldsymbol{\theta}) - \mathcal{I}_Z(\boldsymbol{\theta})\right)\Delta\boldsymbol{\theta} > 0$, then the condition of $\forall \Delta\boldsymbol{\theta} \in \Theta$, $\delta^2 \mathrm{IB}_\beta[p_{\boldsymbol{\theta}}(z|x)] \geq 0$ is equivalent to $\forall \Delta\boldsymbol{\theta} \in \Theta$,

$$\beta \leq \frac{\Delta\boldsymbol{\theta}^T\left(\mathcal{I}_{Z|X}(\boldsymbol{\theta}) - \mathcal{I}_Z(\boldsymbol{\theta})\right)\Delta\boldsymbol{\theta}}{\Delta\boldsymbol{\theta}^T\left(\mathcal{I}_{Z|Y}(\boldsymbol{\theta}) - \mathcal{I}_Z(\boldsymbol{\theta})\right)\Delta\boldsymbol{\theta}}$$

i.e.

$$\beta \leq G_\Theta[p_{\boldsymbol{\theta}}(z|x)] \equiv \inf_{\Delta\boldsymbol{\theta}\in\Theta} \frac{\Delta\boldsymbol{\theta}^T\left(\mathcal{I}_{Z|X}(\boldsymbol{\theta}) - \mathcal{I}_Z(\boldsymbol{\theta})\right)\Delta\boldsymbol{\theta}}{\Delta\boldsymbol{\theta}^T\left(\mathcal{I}_{Z|Y}(\boldsymbol{\theta}) - \mathcal{I}_Z(\boldsymbol{\theta})\right)\Delta\boldsymbol{\theta}} \tag{22}$$

If $\Delta\boldsymbol{\theta}^T\left(\mathcal{I}_{Z|Y}(\boldsymbol{\theta}) - \mathcal{I}_Z(\boldsymbol{\theta})\right)\Delta\boldsymbol{\theta} = 0$, we have that Eq. (21) always holds, which is a looser condition than Eq. (22). Above all, we have that the condition of $\forall \Delta\boldsymbol{\theta} \in \Theta$, $\delta^2 \mathrm{IB}_\beta[p_{\boldsymbol{\theta}}(z|x)]$ is equivalent to $\beta \leq G_\Theta[p_{\boldsymbol{\theta}}(z|x)]$.

Moreover, $(G_\Theta[p_{\boldsymbol{\theta}}(z|x)])^{-1}$ given by Eq. (22) has the format of a generalized Rayleigh quotient $R(A, B; x) \equiv \frac{\Delta\boldsymbol{\theta}^T A \Delta\boldsymbol{\theta}}{\Delta\boldsymbol{\theta}^T B \Delta\boldsymbol{\theta}}$ where $A = \mathcal{I}_{Z|Y}(\boldsymbol{\theta}) - \mathcal{I}_Z(\boldsymbol{\theta})$ and $B = \mathcal{I}_{Z|X}(\boldsymbol{\theta}) - \mathcal{I}_Z(\boldsymbol{\theta})$ are both Hermitian matrices[6], which can be reduced to Rayleigh quotient $R(D, C^T\Delta\boldsymbol{\theta}) = \frac{(C^T\Delta\boldsymbol{\theta})^T D(C^T\Delta\boldsymbol{\theta})}{(C^T\Delta\boldsymbol{\theta})^T(C^T\Delta\boldsymbol{\theta})}$, with the transformation $D = C^{-1}A(C^T)^{-1}$ where $CC^T$ is the Cholesky decomposition of $B = \mathcal{I}_{Z|X}(\boldsymbol{\theta}) - \mathcal{I}_Z(\boldsymbol{\theta})$. Moreover, we have that when $G_\Theta[p_{\boldsymbol{\theta}}(z|x)]$ attains its minimum value, the Reyleigh quotient $R(D, C^T\Delta\boldsymbol{\theta})$ attains its maximum value of $\lambda_{\max}$ with $C^T\Delta\boldsymbol{\theta} = v_{\max}$, i.e. $\Delta\boldsymbol{\theta} = (C^T)^{-1}v_{\max}$, where $\lambda_{\max}$ is the largest eigenvalue of $D$ and $v_{\max}$ the corresponding eigenvector.

$\square$

# D    PROOF OF THEOREM 1

*Proof.* Define

$$T_\beta(\beta') := \inf_{r(z|x)\in\mathcal{Q}_{\mathcal{Z}|\mathcal{X}}} \left[\left(\mathbb{E}_\beta[r^2(z|x)] - \mathbb{E}_\beta[r^2(z)]\right) - \beta'\cdot\left(\mathbb{E}_\beta[r^2(z|y)] - \mathbb{E}_\beta[r^2(z)]\right)\right] \tag{23}$$

where $\mathbb{E}_\beta[\cdot]$ denotes taking expectation w.r.t. the optimal solution $p_\beta^*(x, y, z) = p(x, y)p_\beta^*(z|x)$ at $\beta$. Using Lemma 2.1, we have that the IB phase transition as defined in Definition 3 corresponds to satisfying the following two equations:

$$T_\beta(\beta')|_{\beta'=\beta} \geq 0 \tag{24}$$

$$\lim_{\beta'\to\beta^+} T_\beta(\beta') = 0^- \tag{25}$$

Now we prove that $T_\beta(\beta')$ is continuous at $\beta' = \beta$, i.e. $\forall\varepsilon > 0$, $\exists\delta > 0$ s.t. $\forall\beta \in (\beta - \delta, \beta + \delta)$, we have $|T_\beta(\beta') - T_\beta(\beta)| < \epsilon$.

---

[6]Here all the Fisher information matrices are real symmetric, thus Hermitian.

From Eq. (23), we have $T_\beta(\beta') - T_\beta(\beta) = -(\beta' - \beta) \cdot \left( \mathbb{E}_\beta[r^2(z|y)] - \mathbb{E}_\beta[r^2(z)] \right)$. Since $r(z|x)$ is bounded, i.e. $\exists M > 0$ s.t. $\forall z \in \mathcal{Z}, x \in \mathcal{X}, |r(z|x)| \leq M$, we have

$$\left| \mathbb{E}_\beta \left[ r^2(z|y) \right] \right| = \left| \mathbb{E}_\beta \left[ \left( \mathbb{E}_{x \sim p(x|y,z)} \left[ r(z|x) \right] \right)^2 \right] \right| \leq \left| \mathbb{E}_\beta \left[ \left( \mathbb{E}_{x \sim p(x|y,z)} \left[ M \right] \right)^2 \right] \right| = M^2$$

Similarly, we have

$$\left| \mathbb{E}_\beta \left[ r^2(z) \right] \right| = \left| \mathbb{E}_\beta \left[ \left( \mathbb{E}_{x \sim p(x|z)} \left[ r(z|x) \right] \right)^2 \right] \right| \leq \left| \mathbb{E}_\beta \left[ \left( \mathbb{E}_{x \sim p(x|z)} \left[ M \right] \right)^2 \right] \right| = M^2$$

Hence, $|T_\beta(\beta') - T_\beta(\beta)| = |\beta' - \beta| \left| E_\beta[r^2(z|y)] - E_\beta[r^2(z)] \right| \leq 2|\beta' - \beta|M^2$.

To prove that $T_\beta(\beta')$ is continuous at $\beta' = \beta$, we have $\forall \varepsilon > 0, \exists \delta = \frac{\varepsilon}{2M^2} > 0$, s.t. $\forall \beta' \in (\beta - \delta, \beta + \delta)$, we have

$$|T_\beta(\beta') - T_\beta(\beta)| \leq 2|\beta' - \beta|M^2 < 2\delta M^2 = 2\frac{\varepsilon}{2M^2}M^2 = \varepsilon$$

Hence $T_\beta(\beta')$ is continuous at $\beta' = \beta$.

Combining the continuity of $T_\beta(\beta')$ at $\beta' = \beta$, and Eq. (24) and (25), we have $T_\beta(\beta) = 0$, which is equivalent to $G[p_\beta^*(z|x)] = \beta$ after simple manipulation.

$\square$

# E   INVARIANCE OF $\mathcal{G}[r(z|x); p(z|x)]$ TO ADDITION OF A GLOBAL REPRESENTATION

Here we prove the following lemma:

**Lemma 2.2.** *$\mathcal{G}[r(z|x); p(z|x)]$ defined in Lemma 0.1 is invariant to the transformation $r'(z|x) \leftarrow r(z|x) + s(z)$.*

*Proof.* When we $r(z|x)$ is shifted by a global transformation $r'(z|x) \leftarrow r(z|x) + s(z)$, we have $r'(z) \leftarrow \mathbb{E}_{x \sim p(x|z)}[r(z|x) + s(z)] = \mathbb{E}_{x \sim p(x|z)}[r(z|x)] + s(z)\mathbb{E}_{x \sim p(x|z)}[1] = r(z) + s(z)$, and similarly $r'(z|y) \leftarrow r(z|y) + s(z)$.

The numerator of $\mathcal{G}[r(z|x); p(z|x)]$ is then

$$\mathbb{E}_{x,z \sim p(x,z)} \left[ (r'(z|x))^2 \right] - \mathbb{E}_{z \sim p(z)} \left[ (r'(z))^2 \right]$$
$$= \mathbb{E}_{x,z \sim p(x,z)} \left[ (r(z|x) + s(z))^2 \right] - \mathbb{E}_{z \sim p(z)} \left[ (r(z) + s(z))^2 \right]$$
$$= \left( \mathbb{E}_{x,z \sim p(x,z)} \left[ r^2(z|x) \right] + 2\mathbb{E}_{x,z \sim p(x,z)} \left[ r(z|x)s(z) \right] + \mathbb{E}_{x,z \sim p(x,z)} \left[ s^2(z) \right] \right)$$
$$\quad - \left( \mathbb{E}_{z \sim p(z)} \left[ r^2(z) \right] + 2\mathbb{E}_{z \sim p(z)} \left[ r(z)s(z) \right] + \mathbb{E}_{z \sim p(z)} \left[ s^2(z) \right] \right)$$
$$= \left( \mathbb{E}_{x,z \sim p(x,z)} \left[ r^2(z|x) \right] + 2\mathbb{E}_{z \sim p(z)} \left[ s(z)\mathbb{E}_{x \sim p(x|z)} \left[ r(z|x) \right] \right] + \mathbb{E}_{z \sim p(z)} \left[ s^2(z) \right] \right)$$
$$\quad - \left( \mathbb{E}_{z \sim p(z)} \left[ r^2(z) \right] + 2\mathbb{E}_{z \sim p(z)} \left[ r(z)s(z) \right] + \mathbb{E}_{z \sim p(z)} \left[ s^2(z) \right] \right)$$
$$= \left( \mathbb{E}_{x,z \sim p(x,z)} \left[ r^2(z|x) \right] + 2\mathbb{E}_{z \sim p(z)} \left[ r(z)s(z) \right] + \mathbb{E}_{z \sim p(z)} \left[ s^2(z) \right] \right)$$
$$\quad - \left( \mathbb{E}_{z \sim p(z)} \left[ r^2(z) \right] + 2\mathbb{E}_{z \sim p(z)} \left[ r(z)s(z) \right] + \mathbb{E}_{z \sim p(z)} \left[ s^2(z) \right] \right)$$
$$= \mathbb{E}_{x,z \sim p(x,z)} \left[ r^2(z|x) \right] - \mathbb{E}_{z \sim p(z)} \left[ r^2(z) \right]$$

Symmetrically, we have

$$\mathbb{E}_{y,z \sim p(y,z)} \left[ (r'(z|y))^2 \right] - \mathbb{E}_{z \sim p(z)} \left[ (r'(z))^2 \right] = \mathbb{E}_{y,z \sim p(y,z)} \left[ r^2(z|y) \right] - \mathbb{E}_{z \sim p(z)} \left[ r^2(z) \right]$$

Therefore, $\mathcal{G}[r(z|x); p(z|x)] = \frac{\mathbb{E}_{x,z \sim p(x,z)}[r^2(z|x)] - \mathbb{E}_{z \sim p(z)}[r^2(z)]}{\mathbb{E}_{y,z \sim p(y,z)}[r^2(z|y)] - \mathbb{E}_{z \sim p(z)}[r^2(z)]}$ is invariant to $r'(z|x) \leftarrow r(z|x) + s(z)$.

$\square$

# F    PROOF OF THEOREM 2

*Proof.* Using the condition of the theorem, we have that $\forall r(z|x) \in \mathcal{Q}^0_{\mathcal{Z}|\mathcal{X}}$, there exists $r_1(z|x) \in \mathcal{Q}_{\mathcal{Z}|\mathcal{X}}$ and $s(z) \in \{s : \mathcal{Z} \to \mathbb{R} | s \text{ bounded}\}$ s.t. $r(z|x) = r_1(z|x) + s(z)$. Note that the only difference between $\mathcal{Q}_{\mathcal{Z}|\mathcal{X}}$ and $\mathcal{Q}^{(0)}_{\mathcal{Z}|\mathcal{X}}$ is that $\mathcal{Q}_{\mathcal{Z}|\mathcal{X}}$ requires $\mathbb{E}_{p(z|x)}[r_1(z|x)] = 0$. Using Lemma 2.2, we have

$$\inf_{r(z|x) \in \mathcal{Q}^{(0)}_{\mathcal{Z}|\mathcal{X}}} \mathcal{G}[r(z|x); p(z|x)] = \inf_{r_1(z|x) \in \mathcal{Q}_{\mathcal{Z}|\mathcal{X}}} \mathcal{G}[r_1(z|x); p(z|x)] = G[p(z|x)]$$

where $r(z|x)$ doesn't have the constraint of $\mathbb{E}_{p(z|x)}[\cdot] = 0$.

After dropping the constraint of $\mathbb{E}_{z \sim p(z|x)}[r(z|x)] = 0$, again using Lemma 2.2, we can let $r(z) = \mathbb{E}_{x \sim p(x|z)}[r(z|x)] = 0$ (since we can perform the transformation $r'(z|x) \leftarrow r(z|x) - r(z)$, so that the new $r'(z) \equiv 0$). Now we get a simpler formula for $G[p(z|x)]$, as follows:

$$G[p(z|x)] = \inf_{r(z|x) \in \mathcal{Q}^{(1)}_{\mathcal{Z}|\mathcal{X}}} \frac{\mathbb{E}_{x,z \sim p(x,z)}[r^2(z|x)]}{\mathbb{E}_{y,z \sim p(y,z)}\left[\left(\mathbb{E}_{x \sim p(x|y,z)}[r(z|x)]\right)^2\right]} \tag{26}$$

where $\mathcal{Q}^{(1)}_{\mathcal{Z}|\mathcal{X}} := \{r : \mathcal{X} \times \mathcal{Z} \to \mathbb{R} \,|\, \mathbb{E}_{x \sim p(x|z)}[r(z|x)] = 0, r \text{ bounded}\}$.

From Eq. (26), we can further require that $\mathbb{E}_{x,z \sim p(x,z)}[r^2(z|x)] = 1$. Define

$$\rho_s^2(X, Y; Z) := \sup_{f(X,Z) \in \mathcal{Q}^{(2)}_{\mathcal{Z}|\mathcal{X}}} \mathbb{E}[(\mathbb{E}[f(X,Z)|Y,Z])^2] = \sup_{f(x,z) \in \mathcal{Q}^{(2)}_{\mathcal{Z}|\mathcal{X}}} \mathbb{E}_{y,z \sim p(y,z)}\left[\left(\mathbb{E}_{x \sim p(x|y,z)}[f(x,z)]\right)^2\right] \tag{27}$$

where[7] $\mathcal{Q}^{(2)}_{\mathcal{Z}|\mathcal{X}} := \{r : \mathcal{X} \times \mathcal{Z} \to \mathbb{R} \,|\, \mathbb{E}_{x \sim p(x|z)}[r(z|x)] = 0, \mathbb{E}_{x,z \sim p(x,z)}[r^2(z|x)] = 1, r \text{ bounded}\}$. Comparing with Eq. (26), it immediately follows that

$$G[p(z|x)] = \frac{1}{\rho_s^2(X, Y; Z)}$$

**(i)** We only have to prove that $\rho_s(X, Y; Z) = \rho_r(X, Y; Z)$, where $\rho_r(X, Y; Z)$ is defined in Definition 4.

We have

$$\mathbb{E}[f(X, Z)g(Y, Z)]$$

$$= \int dx dy dz\, p(x, y, z) f(x, z) g(y, z)$$

$$= \int dy dz\, p(y, z) g(y, z) \int dx\, p(x|y, z) f(x, z)$$

$$\equiv \int dy dz\, p(y, z) g(y, z) F(y, z)$$

$$\leq \sqrt{\int dy dz\, p(y, z) g^2(y, z)} \cdot \sqrt{\int dy dz\, p(y, z) F^2(y, z)}$$

where $F(y, z) := \int dx\, p(x|y, z) f(x, z)$. We have used Cauchy-Schwarz inequality, where the equality holds when $g(y, z) = \alpha F(y, z)$ for some $\alpha$. Since $\mathbb{E}[g^2(y, z)] = 1$, we have $\alpha^2 \mathbb{E}[F^2(y, z)] = 1$.

---

[7]In the definition of $\rho_r(X, Y; Z)$, we have used an equivalent format $f(x, z)$ instead of $r(z|x)$.

Taking the supremum of $(\mathbb{E}[f(X,Z)g(Y,Z)])^2$ w.r.t. $f$ and $g$, we have

$$\rho_r^2(X,Y;Z) = \sup_{(f(X,Z),g(Y,Z))\in\mathcal{S}_1} (\mathbb{E}[f(X,Z)g(Y,Z)])^2$$

$$= \sup_{(f(x,z),g(y,z))\in\mathcal{S}_1} \int dydz p(y,z) g^2(y,z) \cdot \int dydz p(y,z) F^2(y,z)$$

$$= \sup_{f(x,z)\in\mathcal{Q}_{\mathcal{Z}|\mathcal{X}}^{(2)}} \int dydz p(y,z) F^2(y,z)$$

$$= \sup_{f(x,z)\in\mathcal{Q}_{\mathcal{Z}|\mathcal{X}}^{(2)}} \int dydz p(y,z) \left(\int dx p(x|y,z) f(x,z)\right)^2$$

$$= \sup_{f(X,Z)\in\mathcal{Q}_{\mathcal{Z}|\mathcal{X}}^{(2)}} \mathbb{E}[(\mathbb{E}[f(X,Z)|Y,Z])^2]$$

$$\equiv \rho_s^2(X,Y;Z)$$

Here $\mathcal{S}_1$ is defined in Definition 4. By definition both $\rho_r(X,Y;Z)$ and $\rho_s(X,Y;Z)$ take non-negative values. Therefore,

$$\rho_s(X,Y;Z) = \rho_r(X,Y;Z) \tag{28}$$

**(ii)** Using the definition of $\rho_r(X,Y;Z)$, we have

$$\rho_r^2(X,Y;Z)$$

$$\equiv \sup_{f(x,z)\in\mathcal{Q}_{\mathcal{Z}|\mathcal{X}}^{(2)}} \int dydz p(y,z) \left(\int dx p(x|y,z) f(x,z)\right)^2$$

$$= \sup_{f(x,z)\in\mathcal{Q}_{\mathcal{Z}|\mathcal{X}}^{(2)}} \int dz p(z) \int dy p(y|z) \left(\int dx p(x|y,z) f(x,z)\right)^2$$

$$\equiv \sup_{f(x,z)\in\mathcal{Q}_{\mathcal{Z}|\mathcal{X}}^{(2)}} \int dz p(z) W[f(x,z)]$$

where $W[f(x,z)] := \int dy p(y|z) \left(\int dx p(x|y,z) f(x,z)\right)^2$.

Denote $c(z) := p(z)\mathbb{E}_{x\sim p(x|z)}[f^2(x,z)]$, we have $\int c(z)dz = \mathbb{E}_{x,z\sim p(x,z)}[f^2(x,z)] = 1$. Then the supremum $\rho_r^2(X,Y;Z) = \sup_{f(x,z)\in\mathcal{Q}_{\mathcal{Z}|\mathcal{X}}^{(2)}} \int dz p(z) W[f(x,z)]$ is equivalent to the following two-stage supremum:

$$\rho_r^2(X,Y;Z) = \sup_{c(z):\int c(z)dz=1} \int dz p(z) \sup_{f(x,z)\in\mathcal{Q}_{\mathcal{Z}|\mathcal{X}}^{(3)}} W[f(x,z)] \tag{29}$$

where $\mathcal{Q}_{\mathcal{Z}|\mathcal{X}}^{(3)} := \{\mathcal{X}\times\mathcal{Z}\to\mathbb{R} \mid \mathbb{E}_{x\sim p(x|z)}[f^2(x,z)] = \frac{c(z)}{p(z)}, \mathbb{E}_{x\sim p(x|z)}[f(x,z)] = 0, f \text{ bounded}\}$
We can think of the inner supremum $\sup_{f(x,z)\in\mathcal{Q}_{\mathcal{Z}|\mathcal{X}}^{(3)}} W[f(x,z)]$ as only w.r.t. $x$, for some given $z$.

Now let's consider another supremum:

$$\sup_{h(x)\in\mathcal{Q}_{\mathcal{X}}^{(h)}} \int dy p(y|z) \left(\int dx p(x|y,z) h(x)\right)^2 \tag{30}$$

where $\mathcal{Q}_{\mathcal{X}}^{(h)} := \{h : \mathcal{X}\to\mathbb{R} \mid \mathbb{E}_{p(x|z)}[h(x)] = 0, \mathbb{E}_{p(x|z)}[h^2(x)] = 1, h \text{ bounded}\}$. Using similar technique in (ii), it is easy to prove that it equals $\rho_m^2(X,Y|Z)$ as defined in Definition 4.

Comparing Eq. (30) and the supremum:

$$\sup_{f(x,z) \in \mathcal{Q}_{\mathcal{Z}|\mathcal{X}}^{(3)}} W[f(x,z)]$$

we see that the only difference is that in the latter $\mathbb{E}_{x \sim p(x|z)}[f^2(x,z)]$ equals $\frac{c(z)}{p(z)}$ instead of 1. Since $W[f(x,z)]$ is a quadratic functional of $f(x,z)$, we have

$$\sup_{f(x,z) \in \mathcal{Q}_{\mathcal{Z}|\mathcal{X}}^{(3)}} W[f(x,z)] = \frac{c(z)}{p(z)} \rho_m^2(X, Y|Z)$$

Therefore,

$$
\begin{aligned}
\rho_r(X, Y; Z) &= \sup_{c(z): \int c(z)dz=1} \int dz\, p(z) \sup_{f(x,z) \in \mathcal{Q}_{\mathcal{Z}|\mathcal{X}}^{(3)}} W[f(x,z)] \\
&= \sup_{c(z): \int c(z)dz=1} \int dz\, p(z) \frac{c(z)}{p(z)} \rho_m^2(X, Y|Z) \\
&= \sup_{c(z): \int c(z)dz=1} \int dz\, c(z) \rho_m^2(X, Y|Z = z) \\
&= \sup_{Z \in \mathcal{Z}} \rho_m^2(X, Y|Z)
\end{aligned}
$$

where in the last equality we have let $c(z)$ have "mass" only on the place where $\rho_m^2(X, Y|Z = z)$ attains supremum w.r.t. $z$.

**(iii)** When $Z$ is a continuous variable, let $f(x,z) = f_X(x)\sqrt{\frac{\delta(z-z_0)}{p(z)}}$, where $\delta(\cdot)$ is the Dirac-delta function, $z_0$ is a parameter, $f_X(x) \in \mathcal{Q}_{\mathcal{X}|\mathcal{Z}}^{(f)}$, with $\mathcal{Q}_{\mathcal{X}|\mathcal{Z}}^{(f)} := \{f_X : \mathcal{X} \to \mathbb{R} \mid f_X \text{ bounded}; \forall Z \in \mathcal{Z} : \mathbb{E}_{X \sim p(X|Z)}[f_X(x)] = 0, \mathbb{E}_{X \sim p(X|Z)}[f_X^2(x)] = 1\}$. We have

$$
\begin{aligned}
\mathbb{E}_{x \sim p(x|z)}[f(x,z)] &= \int p(x|z)f(x,z)dx \\
&= \sqrt{\frac{\delta(z-z_0)}{p(z)}} \int p(x|z)f_X(x)dx \\
&= \sqrt{\frac{\delta(z-z_0)}{p(z)}} \cdot 0 \\
&= 0
\end{aligned}
$$

And

$$
\begin{aligned}
\mathbb{E}[f^2(X, Z)] &= \int p(x,z)f^2(x,z)dxdz \\
&= \int p(x,z)f_X^2(x)\frac{\delta(z-z_0)}{p(z)}dxdz \\
&= \int dz\delta(z-z_0) \int dx p(x|z)f_X^2(x)dx \\
&= \int dz\delta(z-z_0) \cdot 1 \\
&= 1
\end{aligned}
$$

Therefore, such constructed $f(x,z) = f_X(x)\sqrt{\frac{\delta(z-z_0)}{p(z)}} \in \mathcal{Q}_{\mathcal{Z}|\mathcal{X}}^{(2)}$, satisfying the requirement for $\rho_s(X, Y; Z)$ (which equals $\rho_r(X, Y; Z)$ by Eq. 28).

Substituting in the special form of $f(x, z)$ into the expression of $\rho_s(X, Y; Z)$ in Eq. (27), we have

$$\sup_{f(x,z):f(x,z)=f_X(x)\sqrt{\frac{\delta(z-z_0)}{p(z)}}, f_X(x)\in\mathcal{Q}^{(f)}_{\mathcal{X}|\mathcal{Z}}} \int dz p(z) \int dy p(y|z) \left(\int dx p(x|y,z) f(x,z)\right)^2$$

$$= \sup_{f_X(x)\in\mathcal{Q}^{(f)}_{\mathcal{X}|\mathcal{Z}}, z_0\in\mathcal{Z}} \int dz p(z) \int dy p(y|z) \left(\int dx p(x|y,z) f_X(x) \sqrt{\frac{\delta(z-z_0)}{p(z)}}\right)^2$$

$$= \sup_{z_0\in\mathcal{Z}} \int dz p(z) \frac{\delta(z-z_0)}{p(z)} \sup_{f_X(x)\in\mathcal{Q}^{(f)}_{\mathcal{X}|\mathcal{Z}}} \int dy p(y|z) \left(\int dx p(x|y,z) f_X(x)\right)^2$$

$$= \sup_{z_0\in\mathcal{Z}} \int dz \delta(z-z_0) \sup_{f_X(X)\in\mathcal{Q}^{(f)}_{\mathcal{X}|\mathcal{Z}}} \mathbb{E}[(\mathbb{E}[f_X(X)|Y, Z=z])^2|Z=z]$$

$$= \sup_{z_0\in\mathcal{Z}} \int dz \delta(z-z_0) \rho_m^2(X, Y|Z=z)$$

$$= \sup_{z_0\in\mathcal{Z}} \rho_m^2(X, Y|Z=z_0)$$

$$= \sup_{Z\in\mathcal{Z}} \rho_m^2(X, Y|Z)$$

We can identify $\sup_{f_X(X)\in\mathcal{Q}^{(f)}_{\mathcal{X}|\mathcal{Z}}} \mathbb{E}[(\mathbb{E}[f_X(X)|Y, Z=z])^2|Z=z]$ with $\rho_m^2(X, Y|Z=z)$ because $f_X(x)$ satisfies the requirement for conditional maximum correlation that $\mathbb{E}_{p(x|z)}[f_X(x)] = 0$ and $\mathbb{E}_{p(x|z)}[f_X^2(x)] = 1$, for any $z$, and using the same technique in (i), it is straightforward to prove that $\sup_{f_X(X)\in\mathcal{Q}^{(f)}_{\mathcal{X}|\mathcal{Z}}} \mathbb{E}[(\mathbb{E}[f_X(X)|Y, Z=z])^2|Z=z]$ equals the conditional maximum correlation as defined in Definition 4.

Since the conditional maximum correlation can be viewed as the maximum correlation between $X$ and $Y$, where $X, Y \sim p(X, Y|Z)$, using the equality of $(\beta_0[h(x)])^{-1} = \rho_m^2(X; Y)$ (Eq. 7 in Wu et al. (2019)), we can identify the $h(x)$ in $\beta_0[h(x)]$ with the $f_X(X)$ here, and an optimal $f_X^*(X)$ that maximizes $\rho_m^2(X, Y|Z)$ is also an optimal $h^*(x)$ that minimizes $\beta_0[h(x)]$.

**(iv)** For discrete $X$, $Y$ and $Z$ and a given $Z = z$, let $Q_{X,Y|Z} := \left(\frac{p(x,y|z)}{\sqrt{p(x|z)p(y|z)}}\right)_{x,y} = \left(\frac{p(x,y)}{\sqrt{p(x)p(y)}}\sqrt{\frac{p(z|x)}{p(z|y)}}\right)_{x,y}$, we first prove that its second largest singular value is $\rho_m^2(X, Y|Z) = \sup_{(f,g)\in\mathcal{S}_2} \mathbb{E}_{x,y\sim p(x,y|z)}[f(x)g(y)]$ ($\mathcal{S}_2$ is defined in Definition 4).

Let column vectors $u_1 = \sqrt{p(x|z)}$ and $v_1 = \sqrt{p(y|z)}$ (note that $z$ is given and fixed). Also let $u_2 = f(x)\sqrt{p(x|z)}$ and $v_2 = g(y)\sqrt{p(y|z)}$. Denote inner product $\langle u, v\rangle \equiv \sum_i u_i v_i$, and the length of a vector as $||u|| = \sqrt{\langle u, u\rangle}$. We have $||u_1|| = ||v_1|| = 1$ due to the normalization of probability, $||u_2|| = ||v_2|| = 1$ due to $\mathbb{E}_{x\sim p(x|z)}[f^2(x)] = \mathbb{E}_{y\sim p(y|z)}[g^2(y)] = 1$, and $\langle u_1, u_2\rangle = \langle v_1, v_2\rangle = 0$ due to $\mathbb{E}_{x\sim p(x|z)}[f(x)] = \mathbb{E}_{y\sim p(y|z)}[g(y)] = 0$. Furthermore, we have

$$\sup_{(f,g)\in\mathcal{S}_2} \mathbb{E}_{x,y\sim p(x,y|z)}[f(x)g(y)] = \max_{u,v} u^T Q_{X,Y|Z} v$$

which is exactly the second largest singular value $\sigma_2(Z)$ of the matrix $Q_{X,Y|Z}$. Using the result in **(ii)**, we have that $\rho_r(X, Y; Z) = \max_{Z\in\mathcal{Z}} \sigma_2(Z)$.

$\square$

## G  SUBSET SEPARATION AT PHASE TRANSITIONS

In this section we study the behavior of $p(z|x)$ on the phase transitions. We use the same categorical dataset (where $|X| = |Y| = |Z| = 3$ and $p(x)$ is uniform, and $p(y|x)$ is given in Fig. 5). In Fig. 6 we show the $p(z|x)$ on the simplex before and after each phase transition. We see that the first phase transition corresponds to the separation of $x = 2$ (belonging to $y = 2$) w.r.t. $x \in \{0, 1\}$ (belonging to classes $y \in \{0, 1\}$), on the $p(z|x)$ simplex. The second phase transition corresponds to the separation of $x = 0$ with $x = 1$. Therefore, each phase transition corresponds to the ability to distinguish subset of examples, and learning of new classes.

## H  MNIST EXPERIMENT DETAILS

We use the MNIST training examples with class $0, 1, 2, 3$, with a hidden label-noise matrix as given in Fig. 7, based on which at each minibatch we dynamically sample the observed label. We use conditional entropy bottleneck (CEB) (Fischer, 2018) as the variational IB objective, and run multiple independent instances with different the target $\beta$. We jump start learning by started training at $\beta = 100$ for 100 epochs, annealing $\beta$ from 100 down to the target $\beta$ over 600 epochs, and continue to train at the target epoch for another 800 epochs. The encoder is a three-layer neural net, where each hidden layer has 512 neurons and leakyReLU activation, and the last layer has linear activation. The classifier $p(y|z)$ is a 2-layer neural net with a 128-neuron ReLU hidden layer. The backward encoder $p(z|y)$ is also a 2-layer neural net with a 128-neuron ReLU hidden layer. We trained with Adam (Kingma & Welling, 2013) at learning rate of $10^{-3}$, and anneal down with factor $1/(1 + 0.01 \cdot \text{epoch})$. For Alg. 1, for the $f_\theta$ we use the same architecture as the encoder of CEB, and use $|Z| = 50$ in Alg. 1.

## I  CIFAR10 EXPERIMENT DETAILS

We use the same CIFAR10 class confusion matrix provided in Wu et al. (2019) to generate noisy labels with about 20% label noise on average (reproduced in Table 1). We trained $28 \times 1$ Wide ResNet (He et al., 2016; Zagoruyko & Komodakis, 2016) models using the open source implementation from Cubuk et al. (2018) as encoders for the Variational Information Bottleneck (VIB) (Alemi et al., 2016). The 10 dimensional output of the encoder parameterized a mean-field Gaussian with unit covariance. Samples from the encoder were passed to the classifier, a 2 layer MLP. The marginal distributions were mixtures of 500 fully covariate 10-dimensional Gaussians, all parameters of which are trained.

With this standard model, we trained 251 different models at $\beta$ from 1.0 to 6.0 with step size of 0.02. As in Wu et al. (2019), we jump-start learning by annealing $\beta$ from 100 down to the target $\beta$. We do this over the first 4000 steps of training. The models continued to train for another 56,000 gradient steps after that, a total of 600 epochs. We trained with Adam (Kingma & Ba, 2015) at a base learning rate of $10^{-3}$, and reduced the learning rate by a factor of 0.5 at 300, 400, and 500 epochs. The models converged to essentially their final accuracy within 40,000 gradient steps, and then remained stable.

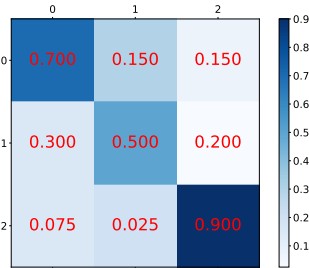

Figure 5: $p(y|x)$ for the categorical dataset in Fig. 2 and Fig. 6. The value in $i^{\text{th}}$ row and $j^{\text{th}}$ column denotes $p(y = j|x = i)$. $p(x)$ is uniform.

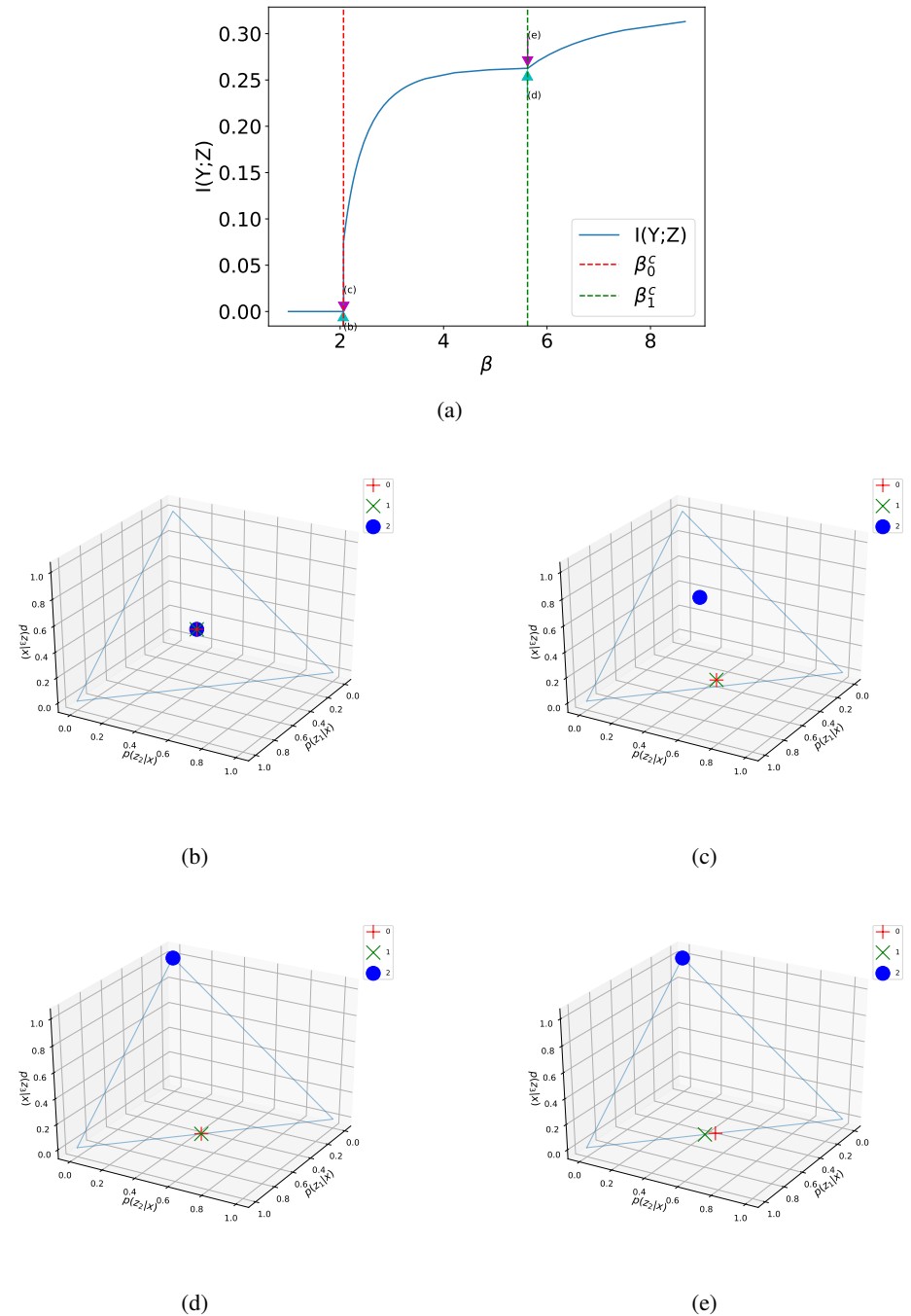

Figure 6: (a) $I(Y; Z)$ vs. $\beta$ for the dataset given in Fig. 5. The phase transitions are marked with vertical dashed line, with $\beta_0^c = 2.065571$ and $\beta_1^c = 5.623333$. (b)-(e) Optimal $p_\beta^*(z|x)$ for four values of $\beta$, i.e. (b) $\beta = 2.060$, (c) $\beta = 2.070$, (d) $\beta = 5.620$ (e) $\beta = 5.625$ (their $\beta$ values are also marked in (a)), where each marker denotes $p(z|x = i)$ for a given $i \in \{0, 1, 2\}$.

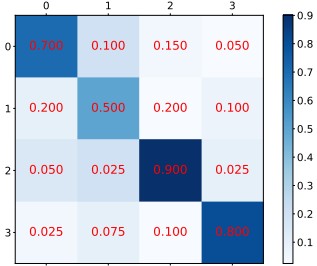

Figure 7: Confusion matrix for MNIST experiment. The value in $i^{\text{th}}$ row and $j^{\text{th}}$ column denotes $p(\tilde{y} = j | y = i)$ for the label noise.

The accuracies reported in Figure 4 are averaged across five passes over the training set. We use $|Z| = 50$ in Alg. 1.

Table 1: Class confusion matrix used in CIFAR10 experiments, reproduced from (Wu et al., 2019). The value in row $i$, column $j$ means for class $i$, the probability of labeling it as class $j$. The mean confusion across the classes is 20%.

|       | Plane   | Auto.   | Bird    | Cat     | Deer    | Dog     | Frog    | Horse   | Ship    | Truck   |
|-------|---------|---------|---------|---------|---------|---------|---------|---------|---------|---------|
| Plane | 0.82232 | 0.00238 | 0.021   | 0.00069 | 0.00108 | 0       | 0.00017 | 0.00019 | 0.1473  | 0.00489 |
| Auto. | 0.00233 | 0.83419 | 0.00009 | 0.00011 | 0       | 0.00001 | 0.00002 | 0       | 0.00946 | 0.15379 |
| Bird  | 0.03139 | 0.00026 | 0.76082 | 0.0095  | 0.07764 | 0.01389 | 0.1031  | 0.00309 | 0.00031 | 0       |
| Cat   | 0.00096 | 0.0001  | 0.00273 | 0.69325 | 0.00557 | 0.28067 | 0.01471 | 0.00191 | 0.00002 | 0.0001  |
| Deer  | 0.00199 | 0       | 0.03866 | 0.00542 | 0.83435 | 0.01273 | 0.02567 | 0.08066 | 0.00052 | 0.00001 |
| Dog   | 0       | 0.00004 | 0.00391 | 0.2498  | 0.00531 | 0.73191 | 0.00477 | 0.00423 | 0.00001 | 0       |
| Frog  | 0.00067 | 0.00008 | 0.06303 | 0.05025 | 0.0337  | 0.00842 | 0.8433  | 0       | 0.00054 | 0       |
| Horse | 0.00157 | 0.00006 | 0.00649 | 0.00295 | 0.13058 | 0.02287 | 0       | 0.83328 | 0.00023 | 0.00196 |
| Ship  | 0.1288  | 0.01668 | 0.00029 | 0.00002 | 0.00164 | 0.00006 | 0.00027 | 0.00017 | 0.83385 | 0.01822 |
| Truck | 0.01007 | 0.15107 | 0       | 0.00015 | 0.00001 | 0.00001 | 0       | 0.00048 | 0.02549 | 0.81273 |

