# OpenReview forum: "Phase Transitions for the Information Bottleneck in Representation Learning"
_ICLR.cc/2020/Conference — Accept (Poster)_

### Official Review · AnonReviewer3 · 2019-10-17
**Official Blind Review #3**

**Rating:** 6

**Review:**

This paper contributes theoretically to the information bottleneck (IB) principle. In particular, the author(s) provided theoretical reasoning on the phase transition phenomenon: when the beta parameter of IB varies, the generalization performance changes in a stepwise manner rather than continuously. The core result is given by theorem 1: the phase transition betas necessarily satisfy an equation, where the LHS is expressed in terms of an optimal perturbation of the encoding function X->Z.

Overall, the reviewer believes that this work is a solid contribution to the IB principle, and should be accepted. Remarkably, this work gives the first formulation on the phase transition of IB, which was an empirical observation. Furthermore, the author(s) give an algorithm to find the transition betas, which agrees with empirical studies on CIFAR and MNIST datasets.

The reasons for the weak acceptance instead of a strong acceptance are explained by the following weaknesses.

As a theoretical contribution, it is important to have good formulations and clear statements. The quality of the mathematical statements is not satisfactory and can be largely improved.

Definition 1: introduce the concept of relative perturbation first. As a definition, it must be self-contained. Therefore introduce the second-order variation first (don't point to the appendix).

Lemma 0.1: inf_r(z|x) \mathcal{G} \ge \beta, is equivalent to \forall r(z|x), \mathcal{G}\ge\beta, the latter is more elegant. What is \mathcal{G}, anyway?

Lemma 0.2:  State the condition on the scale of the $\Delta\theta$. The reviewer suspects that if $\Delta\theta$ is large enough, the statement can be violated.  This lemma feels like a digress. Why the Fisher information is needed for explaining the phase transition?

Theorem 1: the statement is fine. It is better to have all conditions/assumptions listed as this is the core theorem. To be more complete, the authors can discuss the number of roots after the theorem and what happens if there is no root. In general, there have to be a few remarks after theorem 1.

Definition 2: \rho_r(X,Y;Z) is defined with respect to what? For example, state "Given a joined distribution p(X,Y), the representation maximum correlation....". Use a different equivalent symbol for definition, such as ":=".

Theorem 2: Again, \forall{f(x,z)} is with respect to what? In the "if" part of the "if-then" statement, "\forall{f(x,z}}, it can be decomposed as .." is a false statement, change to "If f{x,z} can be decomposed"

In the abstract, avoid citations, and explain X, Y, Z, p(z|x), for completeness.

As the paper is over the recommended length,  the reviewer is asked to be harsher in the assessment.

**Experience Assessment:**

I have read many papers in this area.

**Review Assessment: Checking Correctness Of Derivations And Theory:**

I assessed the sensibility of the derivations and theory.

**Review Assessment: Checking Correctness Of Experiments:**

I assessed the sensibility of the experiments.

**Review Assessment: Thoroughness In Paper Reading:**

I read the paper at least twice and used my best judgement in assessing the paper.

---

> ### Author Response · Authors · 2019-11-11
> **Author Response**
>
> Thank you for your detailed review! We appreciate that you recognize the significance of our work in giving the first formulation on the phase transition of IB, which was an empirical observation. In the revised submission, we have revamped the theorems and proofs according to your suggestions, improving the paper’s rigor, as follows.
>
> Definition 1 IB phase transitions:
> We have added a “Section 3.1 Definitions” at the beginning of Section 3, in which we first state the overall settings and assumption of the paper, then introduce the relative perturbation function (Definition 1) and second variation (Definition 2). Particularly, in introducing the relative perturbation function $r(z|x)$ in Definition 1, we have also introduced $\mathcal{Q}_{\mathcal{Z}|\mathcal{X}}$, which is the set of all valid relative perturbation functions for the probability $p(z|x)$. The definition for IB phase transitions is renamed as Definition 3, in which we have applied the notations in Definition 1, making it more concise. For all the infimum and supremum in the paper, we have made sure that they have well-defined domains.
>
> Lemma 0.1:
> We have now separately stated the definition of $G[p(z|x)]$ and $\mathcal{G}[r(z|x);p(z|x)]$. As regards to your question, although $\forall r(z|x), \mathcal{G}[r(z|x);p(z|x)]\ge\beta$ is more elegant than $\inf\limits_{r(z|x)} \mathcal{G}[r(z|x);p(z|x)] \ge \beta$, the point of Lemma 0.1 is to introduce $G[p(z|x)]:=\inf\limits_{r(z|x)} \mathcal{G}[r(z|x); p(z|x)]$, in preparation for Theorem 1 that gives the condition $G[p^*_\beta(z|x)]=\beta$ for IB phase transitions. Therefore, we use $G[p(z|x)] \ge \beta$ as the condition in Lemma 0.1.
>
> Lemma 0.2:
> We have added the domain for $\Delta\theta$: $\Delta\theta \in \Theta$, where $\Theta$ is the parameter field. There are no requirements for the scale of $\Delta\theta$. We can understand this through an analogy with Taylor expansion in 1D: $f(x_0+h) = f(x_0) + f'(x_0)h + \frac{1}{2}f''(x_0)h^2 + …$. Our $\delta^2 IB[p_{\theta}(z|x)]$ is a quadratic function of $\Delta\theta$, and corresponds to the $\frac{1}{2}f''(x_0)h^2$ term. The requirement of $\forall \Delta\theta\in \Theta$, $\delta^2 IB[p_\theta(z|x)]\ge 0$ corresponds to $\forall h, \frac{1}{2}f''(x_0)h^2\ge 0$, which essentially states that $f''(x_0)\ge 0$ in the *infinitesimal* neighborhood of $x_0$. Note that since $\frac{1}{2}f''(x_0)h^2$ is a quadratic function of $h$, the scale of $h$ does not change the sign of $\frac{1}{2}f''(x_0)h^2$. Thus, $\frac{1}{2}f''(x_0)h^2>0$ is invariant to the scale of $h$. Similarly, $\delta^2 IB[p_\theta(z|x)]\ge 0$ is invariant to the scale of $\Delta\theta$. Therefore, the condition of "$\forall \Delta\theta\in \Theta$, $\delta^2 IB[p_\theta(z|x)]\ge 0$" is expressing the "curvature" in the *infinitesimal* neighborhood of $\theta$, independent of the scale of $\Delta\theta$. We have also clarified this in the proof of the lemma, under Eq. (21). This invariance to the scale of $\Delta\theta$ is carried to the result of the lemma, in that the ratio inside $\inf\limits_{\Delta\theta\in\Theta}$ for $G_{\Theta}[p(z|x)]$ is invariant to the scale of $\Delta\theta$.
>
> The point of the Fisher Information work is to give another way of understanding the phase transitions. The Fisher Information is used extensively to understand machine learning models, so readers who like to think in terms of the Fisher Information should find this result helpful. The reviewer is correct that it is not a required lemma to generate our core results, and we have clarified this in the text.
>
> Theorem 1:
> We have now stated the overall assumption of the paper at the beginning of Section 3, and also in Theorem 1, we have now pointed to Definition 3 for the definition of IB phase transitions, to make it more clear. For the remarks after Theorem 1, in fact, the whole Section 4 is dedicated to the understanding of Theorem 1, which we have now pointed out. Furthermore, at the end of Section 4.1 (Jensen’s inequality), we have added a discussion and conjecture about the number of phase transitions in Theorem 1.
>
> Theorem 2:
> We have improved the statement of the condition in the “if…”, which is a statement about the property of $\mathcal{Q}_{\mathcal{Z}|\mathcal{X}}$, the set of relative perturbation functions, and an expanded set $\mathcal{Q}^{(0)}_{\mathcal{Z}|\mathcal{X}}$, which is $\mathcal{Q}_{\mathcal{Z}|\mathcal{X}}$ without the requirement of $\mathbb{E}_{p(z|x)}[r(z|x)]=0$. We have now made it explicit.
>
> Paper Length:
> With all of the clarifications that we have made, the paper has become slightly longer, although it is still under the strict upper limit of 10 pages. We are open to suggestions on what more we could move to the appendices if the current length does not seem to merit a strong accept.
>
> Abstract:
> For the abstract, we have removed the citations and also explained $X, Y, Z$ and $p(z|x)$, for completeness.
>
> Thank you again for your detailed suggestions!

---

### Official Review · AnonReviewer1 · 2019-10-23
**Official Blind Review #1**

**Rating:** 3

**Review:**

In this paper, the authors studied the phase transition in the information bottleneck, which is defined as the point where the IB loss landscape changes. The authors give a theorem showing the practical condition for IB phase transitions which is related to Fisher information. Then an algorithm finding IB phase transitions are proposed and applied to MNIST and CIFAR10 dataset.

The paper is well-organized. The problem formulation, definition, theorem, algorithm and application form a whole story about finding IB phase transition. However, I found that the motivation of this paper is not very clear. Why do we want to find the IB phase transition accurately by running such a complicated algorithm (Algorithm 1). Usually, the scalar \beta is considered as a hyperparameter. By tuning \beta, one can easily produce Figure 1 and see some phase transition points. One can also tune \beta to learn different classed as shown in Figure 3(b). So designing a complicated coordinate-descent based algorithm to find a parameter seems overkilling the problem.

Moreover, the mathematical part of this paper can be made more accurate. For example, in definition 1, the authors write "\eta << \beta" and "|\epsilon| << 1", where the quantities \eta and \epsilon are used many times in the proof. However,  "<<" is not a rigorous mathematical symbol. The authors should consider rewrite their proof by using limit.

Therefore, I think the motivation and the mathematical quality of this paper can be further improved before getting accepted.

**Experience Assessment:**

I have published one or two papers in this area.

**Review Assessment: Checking Correctness Of Derivations And Theory:**

I assessed the sensibility of the derivations and theory.

**Review Assessment: Checking Correctness Of Experiments:**

I assessed the sensibility of the experiments.

**Review Assessment: Thoroughness In Paper Reading:**

I read the paper at least twice and used my best judgement in assessing the paper.

---

> ### Author Response · Authors · 2019-11-11
> **Author Response**
>
> Thank you for your review. Your perspective has helped us clarify our core theoretical results in particular. We respond to your two major concerns in turn below.
>
> 1. Motivation and importance of Algorithm 1
> We do not consider Algorithm 1 to be the primary contribution of the work. As Reviewer 3 excellently summarizes it, our work contributes theoretically to the information bottleneck (IB) principle, and gives the first formulation on the phase transition of IB, which was an empirical observation. Specifically, our core result is Theorem 1 that gives a theoretical formula for IB phase transitions, which answers the question raised in the introduction “how do they (the phase transitions) depend on the structure of the dataset?”, through our in-depth theoretical analysis of Theorem 1 in Section 4. This question cannot be answered by simply empirically scanning $\beta$. We have improved the "contributions" part in the introduction to make our core contribution more clear.
>
> Algorithm 1, on the other hand, is a natural consequence of our core result Theorem 1, and it allows us to empirically confirm that the theory holds even for non-trivial datasets like MNIST and CIFAR10. We have clarified this in Section 5 where we describe Algorithm 1. However, it is worth comparing the efficiency of Algorithm 1 to other baselines for detecting phase transitions. The naive algorithm of sweeping $\beta$ to identify the phase transitions requires training $K = (\beta_{\text{max}}-\beta_{\text{min}}) / \Delta\beta$ full models to detect phase transitions separated by at least $\Delta\beta$. E.g., for CIFAR10, we trained 251 different VIB models from scratch to sweep out the Pareto-optimal frontier and find likely phase transitions at $\Delta\beta=0.02$ for $\beta$ in $[1.0, 6.0]$. Each of those models took about 36 hours to train on a single GPU. In contrast, with Algorithm 1, we are able to estimate the phase transitions by training exactly 1 full maximum likelihood model (36 hours on a single GPU), and then a handful of small IB models that can all be trained on CPU in a few hours, and the algorithm doesn’t rely on selecting a $\Delta\beta$ step size that would limit the resolution of our search. The difference in computation cost between Algorithm 1 and the naive algorithm of sweeping $\beta$ is massive for even moderately-sized problems like MNIST and CIFAR10. Of course, it is not difficult to do better than the naive algorithm if the goal is to efficiently find phase transitions. For example, iterative refinement algorithms should be able to empirically estimate the phase transitions by training logarithmic numbers of models, rather than the linear number of models in the naive algorithm. If we were trying to write a paper on the best algorithm to empirically find the phase transitions, we would have compared against such approaches. We emphasize again, though, that Algorithm 1 is just one practical consequence of the core theoretical results, rather than the focus of the paper.
>
> 2. Use of limits
> We have revamped the mathematical statements throughout the paper according to your suggestions, which we agree makes a substantial improvement to the core theoretical results. In particular, we have rewritten the theorems and proofs using limits instead of "$\ll$". We have also made other substantial improvements according to Reviewer 3's comments. Please see our responses to Reviewer 3 for more detail.

---

### Official Review · AnonReviewer4 · 2019-11-04
**Official Blind Review #4**

**Rating:** 6

**Review:**

This paper studies the phase transition problem in the information bottleneck (IB) objective and derives a formula for IB phase transitions. Based on the theory developed in the paper, an algorithm is developed to find phase transition points. The interesting observation is phase transition can correspond to learning new class and this paper conjectures in IB for classification, the number of phase transitions is at most C-1, C the number of classes.  This observation deserves to be further explored and may be a key to a deeper understanding of neural networks.

The theory developed connects the IB objective, dataset and the representation, and thorough proofs are given.  The experiments matches the theoretical findings.


**Experience Assessment:**

I do not know much about this area.

**Review Assessment: Checking Correctness Of Derivations And Theory:**

I did not assess the derivations or theory.

**Review Assessment: Checking Correctness Of Experiments:**

I assessed the sensibility of the experiments.

**Review Assessment: Thoroughness In Paper Reading:**

I made a quick assessment of this paper.

---

> ### Author Response · Authors · 2019-11-11
> **Author Response**
>
> Thank you for your review. Your comments made us realize that we had put our conjecture about the number of phase transitions in the experimental section. We have moved it into Section 4.1 on the analysis of Theorem 1 to help make the theoretical basis for the conjecture more clear. We have also made a number of other improvements to the presentation of the core theoretical results, which we describe in our responses to the other two reviewers. We hope that you will find those improvements beneficial as well.

---

### Author Response · Authors · 2019-11-12
**Summary of the revision**

We would like to thank the reviewers for the constructive reviews! We have revised the paper according to the reviewers' comments, and provided detailed responses to each reviewer. A summary of the modification in the revised paper is as follows:

(1) We have rewritten the theorems and proofs using limits instead of "$\ll$"
(2) We have improved the motivation and algorithm parts, making clear that the main contribution of the paper is its theoretical contribution (Theorem 1 and its analysis), with Algorithm 1 a natural consequence.
(3) At the beginning of Section 3, we have stated the overall assumption and settings used throughout the paper.
(4) Before introducing the definition for IB phase transitions, we first define relative perturbation function (Definition 1) and second variation (Definition 2), for a better preparation for the definition of IB phase transitions (now Definition 3).
(5) For all infima and suprema, we have now specified the domain that the argument is with respect to.
(6) For the abstract, we have removed the citations and also explained $X, Y, Z$ and $p(z|x)$, for completeness.
(7) Other changes improving the presentation and rigor of theorems and proofs.

In summary, our work provides the first theoretical formula addressing the Information Bottleneck (IB) phase transitions in the most general setting. Through analysis of the formula, we reveal deep connections between the phase transitions, the structure of the dataset, and the learned representation. Numerical experiments show close matches with the theory (and the resulting algorithm). We believe our work provides novel theoretical insights for the compression vs. prediction tradeoff in IB for representation learning, and our technique may also be inspirational and applicable for the understanding of other “trade-off” objectives, where the model’s ability to predict is balanced against some measure of complexity.

---

### Decision · Program_Chairs · 2019-12-19

**Decision:**

Accept (Poster)

**Comment:**

This submission presents a theoretical study of phase transitions in IB: adjusting the IB parameter leads to step-wise behaviour of the prediction.  Quoting R3: “The core result is given by theorem 1: the phase transition betas necessarily satisfy an equation, where the LHS is expressed in terms of an optimal perturbation of the encoding function X->Z.”
This paper received a borderline review and two votes for weak accept.  The main comment for the borderline review was about the rigor of a proof and the use of << symbols.  The authors have updated the proof using limits as requested, addressing this primary concern.  On the balance, the paper makes a strong contribution to understanding an important learning setting and a contribution to theoretical understanding of the behavior of information bottleneck predictors.